# Endemicity of *Paragonimus* and paragonimiasis in Sub-Saharan Africa: A systematic review and mapping reveals stability of transmission in endemic foci for a multi-host parasite system

**Muriel Rabone**[1,2]*, **Joris Wiethase**[3], **Paul F. Clark**[1], **David Rollinson**[1,2], **Neil Cumberlidge**[4], **Aidan M. Emery**[1,2]

1 Department of Life Sciences, Natural History Museum, Cromwell Road, London, United Kingdom,
2 London Centre for Neglected Tropical Disease Research, School of Public Health, Faculty of Medicine, Imperial College London, Norfolk Place, London, United Kingdom, 3 Department of Biology, University of York, Wentworth Way, York, United Kingdom, 4 Department of Biology, Northern Michigan University, Marquette, Michigan, United States of America

* m.rabone@nhm.ac.uk

## Abstract

Paragonimiasis is caused by zoonotic trematodes of *Paragonimus* spp., found in Asia, the Americas and Africa, particularly in tropical regions. These parasites have a complex, multi-host life cycle, with mammalian definitive hosts and larval stages cycling through two intermediate hosts (snails and freshwater decapod crustaceans). In Africa, paragonimiasis is particularly neglected, and remains the only human parasitic disease without a fully characterised life cycle. However paragonimiasis has potentially significant impacts on public health in Africa, and prevalence has likely been underestimated through under-reporting and misdiagnosis as tuberculosis due to a similar clinical presentation. We identified the need to synthesise current knowledge and map endemic foci for African *Paragonimus* spp. together with *Poikilorchis congolensis*, a rare, taxonomically distant trematode with a similar distribution and morphology. We present the first systematic review of the literature relating to African paragonimiasis, combined with mapping of all reported occurrences of *Paragonimus* spp. throughout Africa, from the 1910s to the present. In human surveys, numerous reports of significant recent transmission in Southeast Nigeria were uncovered, with high prevalence and intensity of infection. Overall prevalence was significantly higher for *P. uterobilateralis* compared to *P. africanus* across studies. The potential endemicity of *P. africanus* in Côte d'Ivoire is also reported. In freshwater crab intermediate hosts, differences in prevalence and intensity of either *P. uterobilateralis* or *P. africanus* were evident across genera and species, suggesting differences in susceptibility. Mapping showed temporal stability of endemic foci, with the majority of known occurrences of *Paragonimus* found in the rainforest zone of West and Central Africa, but with several outliers elsewhere on the continent. This suggests substantial under sampling and localised infection where potential host distributions overlap. Our review highlights the urgent need for increased sampling in active

**Data Availability Statement:** All relevant data are within the manuscript and its Supporting Information files.

**Funding:** The author(s) received no specific funding for this work.

**Competing interests:** The authors have declared that no competing interests exist.

disease foci in Africa, particularly using molecular analysis to fully characterise *Paragonimus* species and their hosts.

## Author summary

Paragonimiasis is a lung disease caused by food-borne zoonotic trematodes of *Paragonimus* spp., multi-host parasites whose complex life cycle includes primary (snail), secondary (freshwater decapod crustacea), and mammalian definitive hosts. In Africa, paragonimiasis is a particularly neglected disease, where prevalence has likely been underestimated through under reporting and misdiagnosis as tuberculosis; and it remains the only human parasite without a fully known life-cycle. We present the first systematic review of paragonimiasis in Africa, including 143 publications. In human studies, we uncovered substantial recent transmission in Southeast Nigeria, and recent transmission also in Côte d'Ivoire and Cameroon in endemic foci; significantly higher prevalence of *P. uterobilateralis* than *P. africanus*; and evidence supporting the existence of *P. africanus* in Côte d'Ivoire. In freshwater crab intermediate hosts, prevalence and intensity of either *P. uterobilateralis* or *P. africanus* varied across genera and species, suggesting differences in susceptibility. Mapping revealed evidence of temporal stability at endemic foci in rainforest regions; and widespread outliers. Numerous reports of significant recent transmission, particularly in Southeast Nigeria should be heeded by the international NTD research community. Increased studies are urgently needed to ascertain the real distribution and diversity of *Paragonimus* species in Africa.

## Introduction

Paragonimiasis is caused by food-borne zoonotic trematodes of *Paragonimus* spp., multi-host parasites whose complex life cycle includes a primary (or first intermediate) snail host; a secondary (or second intermediate) decapod crustacean host, and a mammalian definitive host. Human paragonimiasis has a global distribution in the Americas, Africa, and Asia, particularly in tropical regions [1,2]. An estimated 23 million people are infected worldwide, with the greatest disease burden in Asia [1–3]. The taxonomy of *Paragonimus* is problematic and in need of revision, but currently there are over 50 recognised species, 7 of which cause disease in humans [1,2]. Paragonimiasis in humans is generally caused by the consumption of incompletely cooked freshwater crabs that harbour the infective metacercarial larval stages of the parasite; but it can also be acquired by eating uncooked paratenic hosts, (such as cases from eating wild boar in Japan) which are infected with juvenile worms in development stasis [1,4]. *Paragonimus* is a euryxenous parasite with numerous mammalian reservoir hosts, including for example mongooses, civets, primates and domestic dogs and cats in Africa [5–8]. The parasite also develops in a broad range of molluscan first intermediate hosts, such as Rissooidea and Cerithioidea in Asia [9], and numerous crustacean second intermediate hosts, mostly species of brachyuran crab found in freshwater habitats, some of which are semi-terrestrial [10]. In terms of the clinical picture, paragonimiasis is primarily a disease of the lungs, where adult worms lodge in lung tissue, although ectopic infections are relatively common in some Asian species [2]. The main clinical symptoms include persistent cough, hemoptysis, and abdominal or chest pain [3]. With often overlapping symptoms, there is the potential for diagnostic confusion with tuberculosis (TB); [3,11].

In Africa, paragonimiasis-like lung infections were first reported in the 1920s [12,13], and subsequently recorded only occasionally in the following 3 decades [14–18]. The first descriptions of African species of *Paragonimus* were published in 1965, for *Paragonimus africanus* [7] and *Paragonimus uterobilateralis* [8]. Paragonimiasis in Africa came to widespread attention with an outbreak in Southeast Nigeria in the late 1960s, stemming from food shortages and concomitant consumption of freshwater crabs as a result of the Nigerian Civil war of 1967–1970 [19–21]. In more recent times, paragonimiasis in Africa has received remarkably little attention from the international research community, despite several recent studies reporting significant transmission within Nigeria in particular [22–27]; and also Cameroon [28,29]. Paragonimiasis in Africa remains the only human parasite without a fully characterised life cycle, with the first intermediate snail hosts of the African species of *Paragonimus* unconfirmed, and the miracidia, rediae and cercarial life stages undescribed. Suspected host snail species co-occurring in known crab host habitats include: Achatinidae in Cameroon [30,31]; *Potadoma freethii* in Cameroon [7] and Nigeria [26,32]; *Melania* in Nigeria [33]; and *P. sanctipauli*, *Afropomus balanoidea* [34], and *Homorus* (*Striosubulina*) *striateila* (Achatinidae) in Liberia [35]. The latter studies in Liberia attempted experimental infections, but the candidate *Paragonimus*-like cercariae failed to infect known freshwater crab hosts (*Liberonautes*) collected at the disease foci [34,36]. The second intermediate hosts of *Paragonimus* in Africa are better known, with 8 species of freshwater crabs in 2 genera, *Sudanonautes* and *Liberonautes*, implicated as secondary hosts [10]. However, current taxonomic revisions involving host species may alter the generic assignments of some taxa.

Four species of *Paragonimus* have been recorded from Africa to date: *P. uterobilateralis*, *P. africanus*, *P. gondwanensis*, and *P. kerberti* [7,8,37,38]. All species descriptions are based on morphology only, none have yet been characterised with molecular data, and are therefore currently distinguished by the morphological characters of egg and metacercariae. Only three sequences of one species have been published on GenBank to date, 5.8S, 28S rRNA and ITS2 from *P. africanus* [29,39]. Given that the two species *P. gondwanensis* and *P. kerberti* were described as recently as 2014 and 2015 without molecular data [37,38], and that morphological plasticity is well known in *Paragonimus* [1], they should be regarded as unconfirmed. Aka et al. [40] suggested there may be undescribed species of *Paragonimus* in West and Central Africa, but the lack of molecular data is hampering the ability to answer this question, as well as the study of the life cycle and distribution of these parasites more generally.

Another trematode has been reported from *Paragonimus* endemic regions: *Poikilorchis congolensis* (also known as *Achillurbainia congolensis*). This trematode is the monotypic species of *Poikilorchis* and assigned to the family Achillubainiidae Dolfus, 1939, which has only one other species, *Achillurbainia nouveli* Dolfus, 1939 [41]. While *P. congolensis* is taxonomically distant, early publications show diagnostic confusion with *Paragonimus*, with similar egg morphology and overlapping endemism. However, symptomology of a retroauricular or neck cyst [41] never reported in paragonimiasis, and egg negative sputum, appear consistent with infection of *P. congolensis* only, not *Paragonimus* [42,43].

Very few attempts at spatial characterisation of paragonimiasis in Africa have been made since the mapping of freshwater crab hosts in endemic foci in Nigeria and Cameroon by Voelker & Sachs [44], despite the need to map endemic foci noted as early as 1972 by Nwokolo [20]. Mapping is essential for highly focal diseases, for example recent studies illustrate the temporal stability of endemic foci in schistosomiasis [45]. Foci of paragonimiasis may be similarly stable over time, given the likelihood of the relatively long persistence of mature metacercariae in freshwater crab hosts, and of adult flukes in the lungs of mammalian definitive hosts [1,4]. Although recent reviews have discussed the need to address the extensive knowledge gaps of paragonimiasis in Africa and update the true public health significance of the disease

[46–49], no systematic review or meta-analysis has been performed to date. Consequently, the aims of the current study are: to undertake a systematic review of paragonimiasis in Africa (including reports of the distantly-related trematode, *Poikilorchis congolensis* with diagnostic overlap with *Paragonimus*); to synthesise current knowledge and to map endemic foci. The specific aims of the present study are: 1. to ascertain the level of recent disease transmission; 2. to map reported confirmed occurrences of both *Paragonimus* and the freshwater crab intermediate hosts; and 3. to ultimately establish both the historical and recent distribution of this most neglected tropical disease throughout Africa.

## Materials and methods

### Ethics statement

The data included in this study were anonymised aggregated data from previously published studies only.

### Study protocol

A systematic review was undertaken according to Preferred Reporting Items for Systematic Reviews and Meta-Analyses (PRISMA) guidelines and checklist [50,S1 File]. The *a priori* review question was as follows:

*What is the evidence, extent and distribution of recent/ongoing transmission of paragonimiasis in Africa?*

### Eligibility criteria

Peer reviewed publications on African *Paragonimus* spp./paragonimiasis were included in all languages (included Dutch, English, French, German, Italian, and Japanese).

### Information sources

Online databases were searched through the Natural History Museum (NHM) London library, including: Google Scholar, African Journals Online, PubMed, ScienceDirect, and Web of Science. Publications not openly accessible were requested through interlibrary loans, and ones unavailable by those means were sourced directly in some instances by contacting authors/ researchers in the field (David Aka, David Blair, and Chris Appleton). All citations were comprehensively checked, and any additional publications included. References of all papers were also systematically checked and added if they had not previously been identified through database searches.

### Period

The review was carried out from November 2018-June 2020 by authors MER and JW. The date of last search was the 20th June, 2020.

### Search strategies

Search terms included: paragonimiasis, *Paragonimus*, *Paragonimus africanus/uterobilateralis/ gondwanensis/kerberti*, Africa, for all databases. Additional searches including these terms and the following were made in combination searches: all country names for West and Central Africa (Nigeria, Cameroon, Côte d'Ivoire, Gabon, Equatorial Guinea, Ghana, Liberia, Guinea, Senegal, Central African Republic, Democratic Republic of the Congo/DRC, Republic of the Congo, Central African Republic, Benin, Togo, Sierra Leone, Senegal, Gambia and Burkina

Faso); names of endemic regions, (Cross River, Imo River); prior country names to ensure historical records were found (Haute-Volta, Zaire); relevant French terms (paragonimoses and distomatoses), and intermediate host crab genera names: *Sudanonautes*, *Liberonautes*, *Potamonautes* and *Potamonemus*.

Following initial investigations, an additional review was undertaken for *Poikilorchis congolensis* (also known as *Achillurbainia congolensis*), given the confusion in identification with *Paragonimus* evident in the literature. Cases reporting a retroauricular or neck cyst and egg-negative sputum were assessed to be consistent with infection of *P. congolensis* only, not *Paragonimus*, as evident in the literature [41–43]. Search terms were as above, minus the crab genera names and replacing *Paragonimus* spp. with *Poikilorchis/congolensis*.

## Study selection

Criteria for inclusion were publications on African paragonimiasis only, those addressing Asian or American paragonimiasis/*Paragonimus* species were excluded. All original research, including non-survey based works, for example case reports, experimental studies or questionnaire-only based studies were included. Any publications that did not contain new data, i.e., analysing the findings of an earlier study, were excluded. All papers meeting selection criteria were read in full by MER and JW, apart those in German, read solely by JW. Other papers not in English were translated in Google Translate. The sole paper in Japanese, which also had relevant sections in English, was checked by a native speaker. Reviews were excluded, but those on paragonimiasis in Africa were read in full and references checked against the main publication list and any new references added [46–49,51,52].

## Data collection process

Authors MER and JW extracted data from the publications and cross checked a proportion of the other's data records. A database was created in Microsoft Excel 365 (S1–S5 Tables). Alongside key publication details, data recorded included the following: country; year of study; category (e.g., case report, survey, and for latter, scope of survey = humans/freshwater crabs/freshwater snails or combinations of); *Paragonimus* species reported; a brief description of the study and key findings; what the paragonimiasis infection was described from/how it was diagnosed, e.g., eggs in sputum (S1 Table). Where present, clinical findings were also recorded, e.g., symptoms observed, and details for any other diseases screened for (e.g., TB, and soil transmitted helminths). Other data recorded included: prevalence measurements for any hosts surveyed; egg size measurements where reported; and parasite counts for analysing intensity of infection (egg or metacercariae numbers per host or adult worm counts for wild-caught mammals), as overall averages and individual counts where available. Additional tables for prevalence, intensity and egg-size measurement data were created to allow for the analysis and recording of individual measurements (S2–S4 Tables). In many cases study period was not reported in the paper, here it was estimated as 2 years prior to publication. In analyses of dates, decade was used to mitigate potential estimation bias around sampling period. For human cases, the period or year when symptoms first appeared was noted (if recorded). If locality maps or photographs of collected larval stages were included, this was recorded. Additional data were sourced directly from authors if required for analysis (see Acknowledgements). All publications, both included and excluded from analysis, are listed in S1 Table.

## Methodological assessment

**Identification.** Identification of *Paragonimus* in the literature was generally undertaken by microscopy, i.e., the microscopic identification of eggs or metacercariae. As only 4 studies

used molecular analysis and/or serology [29,39,53,54], here identification was therefore accepted on microscopy only. This criterion was applied even if the result was contrary to the diagnosis in the paper, e.g., a positive case based on serology results or symptoms only was recorded here as unconfirmed. This assessment was made for consistency and to ensure that only current infections were identified. In cases where *Paragonimus* eggs were isolated from stool only, any available photographs were assessed, and evidence of positive identification of other parasite species checked to rule out the possibility of misidentification of another trematode.

**Assessment of surveys.** In human surveys, publications were excluded from further analysis if the study population was selected from people pre-determined as likely infection candidates, for example hospital surveys of pulmonary clinic patients. A publication was also excluded if the total study population was not given. In some papers, study populations were identified and subsequently only those individuals presenting symptoms were recruited for the survey. These publications were included as the original study population total was also given. Both cross-sectional (community-wide) and school-based surveys were included. For mammal surveys also, studies were excluded if total study population was not provided.

**Data analysis and summary statistics.** All data were analysed in R, version 3.5.3 (2019-03-11) "Great Truth". The majority of the literature in the systematic review was highly heterogeneous and a qualitative review was considered the best approach for analysis. A subset of publications was also selected for a comparison of prevalence and intensity across studies, separately for each host category. Averages and standard deviation values were computed in R, using the tidyverse package [55]. Weighted means were computed, for example, egg measurement averages in a given study weighted by total eggs measured; average intensity, weighted by total number of infected hosts (per host-category) per study, and in prevalence, by total surveyed for the study (Table 1). The statistical significance of differences across studies between groups, for example in prevalence and intensity of infected hosts were tested with a Wilcoxon rank sum test, which does not assume a normal distribution, given the high heterogeneity and variance among studies. All statistical tests were conducted with the significance level $\alpha = 0.05$ for rejecting null hypotheses.

**Prevalence.** Prevalence levels were recorded for all hosts: human/other mammals, and intermediate freshwater crab hosts. Prevalence for crabs was calculated by species, excluding *Callinectes marginatus*, regarded as an unconfirmed host [56;NC, pers. comm.]. Prevalence differences by demographics (sex and age) were also examined. For sex, both overall total and proportion of female/male patients positive were recorded, and whether any difference was significant, if reported. For age data, the defined age categories, and proportion of prevalence per category were recorded (S2 Table).

**Intensity.** Range and average for intensity of infection (egg/metacercariae count per host) were recorded, and the average over the study calculated if not reported (S3 Table). Generally, eggs were recorded per 5 ml of sputum (e/5g), or eggs per gram of stool (epg). In one case where egg counts were reported for 1ml sputum samples, this was recorded and counts adjusted to per 5ml to allow for comparability. In another publication, sputum samples of 1-5ml were collected, and this study was excluded from analysis. The sputum and stool sample preparation methods for human and mammal surveys, details on crab dissection and screening for infection, and egg/metacercariae counts per host were recorded. If multiple samples were collected per host and the average reported, this was recorded also. If intensity was reported only by level (in 5ml sputum), these totals were not included as direct comparison was not possible with average egg count totals. Egg count measurements from stool (in addition to sputum) as reported in 3 studies were excluded for consistency. No experimental infections were included in the subset analysis for intensity or prevalence, but measurements of

**Table 1. Weighted averages: Prevalence and intensity of infection, by host category (human, other mammal or crab).**

| Countries | Species | Cat. | Host | Prev. | SD | Inf. | N | Refs | Int. | SD | Inf. | N | Refs | Mammals* |
|---|---|---|---|---|---|---|---|---|---|---|---|---|---|---|
| C, EG | *P. africanus* | H | | 2.9 | 3.2 | 263 | 8978 | 8 | | | | | | |
| L, N | *P. uterobilateralis* | | | 12.4 | 6.8 | 1670 | 13443 | 12 | 108.5 | 84.9 | 1716 | 10477 | 9 | |
| C, G, N | *Paragonimus* sp. | | | 8.7 | 7.8 | 992 | 11415 | 11 | 10.8 | 53.0 | 693 | 4700 | 4 | |
| C | *P. africanus* | C | *S. africanus* | 42.9 | 28.0 | 103 | 240 | 4 | 54.6 | 85.7 | 106 | 214 | 2 | |
| C | *P. africanus* | | *S. aubryi* | 6.3 | - | 1 | 16 | 1 | 3.8 | - | 1 | 16 | 1 | |
| C | *P. africanus* | | *S. pelli* | 1.5 | - | 1 | 66 | 1 | 24.3 | - | 1 | 66 | 1 | |
| C | *P. kerberti* | | *S. africanus* | 32.9 | - | 162 | 493 | 1 | | | | | | |
| L | *P. uterobilateralis* | | *L. chaperi* | 2.3 | - | 1 | 43 | 1 | 2.0 | - | 1 | 40 | 1 | |
| L | *P. uterobilateralis* | | *L. latidactylus* | 24.4 | 36.1 | 87 | 357 | 3 | 117.1 | 286.5 | 85 | 191 | 3 | |
| L | *P. uterobilateralis* | | *L. nanoides* | 0.5 | - | 4 | 763 | 1 | 1.0 | - | 4 | 763 | 1 | |
| N | *P. uterobilateralis* | | *S. sp.* | 5.5 | 2.1 | 235 | 4249 | 2 | | | | | | |
| C, G, N | *P. uterobilateralis* | | *S. africanus* | 5.9 | 41.7 | 156 | 2634 | 5 | 6.0 | 9.8 | 34 | 189 | 2 | |
| C, G, N | *P. uterobilateralis* | | *S. aubryi* | 64.8 | 42.8 | 57 | 88 | 3 | 3.5 | 0.5 | 12 | 43 | 2 | |
| C | *P. uterobilateralis* | | *S. pelli* | 3.0 | - | 2 | 66 | 1 | 0.8 | - | 2 | 66 | 1 | |
| Co | *Paragonimus* sp. | | *L. latidactylus* | 20.0 | - | 6 | 30 | 1 | | | | | | |
| C, N | *Paragonimus* sp. | | *S. africanus* | 10.4 | 20.2 | 342 | 3304 | 4 | 17.5 | - | 94 | 207 | 1 | |
| C | *Paragonimus* sp. | | *S. aubryi* | 10.0 | - | 5 | 50 | 1 | 2.0 | - | 5 | 59 | 1 | |
| C | *Paragonimus* sp. | | *S. granulatus* | 8.1 | - | 3 | 37 | 1 | 4.0 | - | 3 | 37 | 1 | |
| C | *Paragonimus* sp. | | *S. pelli* | 0.0 | - | 0 | 11 | 1 | | | | | | |
| C | *P. africanus* | M | Feliformia | 58.8 | 33.8 | 10 | 17 | 3 | | | | | | ***Civettictis civetta, Crossarchus obscurus, Nandinia binotata*** |
| C, N | *P. africanus* | | Primates | 39.1 | 13.0 | 25 | 64 | 3 | 96.7 | - | 21 | 49 | 1 | ***Cercocebus torquatus*** (& Int.), ***Mandrillus leucophaeus, Perodicticus potto*** |
| L | *P. uterobilateralis* | | domestic | 33.3 | 7.9 | 3 | 9 | 1 | | | | | | **dogs** |
| L | *P. uterobilateralis* | | Eulipotyphla | 12.5 | - | 1 | 8 | 1 | | | | | | ***Crocidura flavescens*** |
| L, N | *P. uterobilateralis* | | Feliformia | 77.8 | 29.1 | 35 | 45 | 4 | 17.0 | - | 2 | 5 | 1 | ***Civettictis civetta*** (& Int.), ***Crossarchus obscurus*** |
| L | *P. uterobilateralis* | | Rodentia | 16.7 | - | 1 | 6 | 1 | | | | | | ***Malacomys edwarsi*** |
| C, Co, Gh, N | *Paragonimus* sp. | | domestic | 0.5 | 7.9 | 2 | 387 | 4 | | | | | | **dogs, pigs**, (cats) |
| N | *Paragonimus* sp. | | Eulipotyphla | 0.0 | - | 0 | 24 | 1 | | | | | | (*Crocidura flavescens*) |
| C, Co, G, N | *Paragonimus* sp. | | Feliformia | 9.5 | 34.5 | 8 | 84 | 5 | | | | | | ***Atilax paludinosus, Civettictis civetta, Crossarchus obscurus, Ichneumia albicauda,*** (*Nandinia binotata*), (*Genette maculata*) |
| C, T | *Paragonimus* sp. | | Primates | 14.1 | 7.4 | 47 | 333 | 4 | 47.0 | - | 23 | 161 | 1 | ***Papio anubis*** (& Int.) (*Cercocebus galeritus*), (*Cercopithecus aethiops*), (*Cercopithecus mona*), (*Galagoides demidovii*) |
| C, G, N | *Paragonimus* sp. | | Rodentia | 0.0 | 0.0 | 0 | 55 | 3 | | | | | | (*Atherurus africanus*), (*Cricetomys emini*), (*Hystrix cristat*), (*Malacomys edwarsi*), (*Cricetomys gambianus*) |

Key: Countries: C = Cameroon, G = Gabon, L = Liberia, EG = Equatorial Guinea, Co = Côte d'Ivoire, Gh = Ghana, N = Nigeria, T = Tanzania, Cat. = Host Category: H: Human, C: Freshwater Crab, M: Mammal, Inf. = total infected, Int. = intensity, Host = Host species (or category for mammals), Prev. = prevalence, SD = standard deviation, N = total surveyed, left of bar: totals for prevalence, right of bar, totals for intensity data, Refs = total number of publications, Mammals* = species of mammal surveyed, those with individuals positive for infection in bold, those not infected, in brackets.

eggs isolated from experimental hosts were included in egg measurement analysis and the host details recorded (S4 Table).

## Mapping

**Data collection.** Occurrence data (each unique sampling event; site and timepoint) were collected and assessed for mapping (S5 Table). All occurrences were recorded individually from each publication meeting criteria for inclusion in the study, e.g., a case report confirmed with evidence of eggs in sputum. No experimental studies were included, unless field collected data were also present. Locality data were also extracted from Table 2 of Cumberlidge et al., 2018 [46] and validated against the literature. Occurrence data collected included: year/s study was undertaken; country; region (verbatim) site name; verbatim site coordinates (if present); site type (village, school, hospital, crab collection site: river/stream or crab market, mammal survey site: forest); *Paragonimus* species, life stage, host species/taxonomy, totals of host surveyed and number infected per site, if recorded. Contextual notes on the occurrence, the locality and the infection in question were also made. Absence data for *Paragonimus* spp. were recorded if present, e.g., if a crab survey did not find infected crabs at a given site. Additional spatial data sources included river and waterway shapefiles, downloaded from DIVA-GIS [57]; and crab intermediate host occurrences. These geolocated occurrence records for the freshwater crab second intermediate hosts in West Africa are based on empirically collected data from field research carried out by NC from 1980 to 1984 in Nigeria and Cameroon, and from 1988 to 1989 in Liberia (S6 Table).

**Inclusion criteria.** Records with a confirmed life stage of *Paragonimus* spp. (e.g., eggs in sputum or metacercariae in crabs) were mapped, in the absence of other diagnostic markers for the disease. Six publications not in the systematic review were included in mapping, either where locality data were more comprehensive than a duplicate publication that was included

**Table 2. Egg measurements in μm across studies were reported, for human host, most isolated from sputum samples, some from stool, mammals, from stool or isolated from adult worms.**

| species | stage | host | Len. wm | Len. av | Len. sd | Wid. wm | Wid. av | Wid. sd | Lmin wm | Lmin av | Lmax wm | Lmax av | Wmin wm | Wmin av | Wmax wm | Wmax av | tot meas | Refs wm | Refs all |
|---|---|---|---|---|---|---|---|---|---|---|---|---|---|---|---|---|---|---|
| *P. africanus* | egg | H | 91.8 | 91.7 | 3.4 | 49.7 | 49.8 | 1.9 | 75.0 | 77.1 | 112.6 | 107.5 | 42.3 | 43.9 | 42.3 | 57.3 | 1220 | 6 | 7 |
| *P. africanus* | egg | M | 92.9 | 91.8 | 3.0 | 49.4 | 49.0 | 1.7 | 77.3 | 77.1 | 105.3 | 106.9 | 41.9 | 42.8 | 41.9 | 56.9 | 1252 | 5 | 6 |
| *P. africanus* | meta | C | 564.0 | 558.3 | 228.8 | | | | 420.0 | 387.0 | 540.0 | 499.3 | | | | | 90 | 2 | 3 |
| *P. uterobilateralis* | egg | H | 68.8 | 70.0 | 4.3 | 40.4 | 41.2 | 2.2 | 53.0 | 57.1 | 86.2 | 82.7 | 34.5 | 35.7 | 34.5 | 47.3 | 2779 | 5 | 8 |
| *P. uterobilateralis* | egg | M | 69.4 | 68.9 | 3.5 | 42.7 | 41.9 | 1.9 | 62.3 | 62.0 | 75.7 | 75.2 | 39.3 | 38.6 | 39.3 | 44.9 | 380 | 4 | 5 |
| *P. uterobilateralis* | meta | C | 701.6 | 734.4 | 119.4 | 304.1 | 314.5 | 30.4 | 449.8 | 550.0 | 929.7 | 953.3 | 244.0 | 244.0 | 244.0 | 396.0 | 128 | 3 | 4 |
| *Paragonimus sp.* | egg | H | 95.7 | 92.2 | 13.8 | 58.8 | 56.5 | 13.5 | 78.8 | 78.0 | 109.5 | 103.8 | 48.0 | 45.4 | 48.0 | 62.4 | 155 | 6 | 13 |
| *Paragonimus sp.* | egg | M | 91.7 | 89.8 | 7.5 | 54.9 | 53.0 | 5.3 | 73.5 | 71.4 | 106.1 | 101.8 | 44.4 | 43.1 | 44.4 | 60.5 | 317 | 3 | 7 |
| *Paragonimus sp.* | meta | C | 468.3 | 558.0 | 144.2 | | | | | | | | | | | | 166 | 2 | 2 |
| *Poikilorchis congolensis* | egg | H | | 65.0 | 7.4 | 43.3 | 43.3 | 8.2 | | | | | | | | | | | 6 |

Key: Len. wm = weighted mean length (weighted by total number of eggs measured), Len. av = overall/average length (all studies, non-weighted, including studies where total of eggs measured were not reported), Len. sd = standard deviation of average length, Wid. wm = weighted mean width, Wid. av = average width across studies, W sd = standard deviation of average width, W sd = standard deviation of width, Lmin wm = weighted mean of minimum recorded length, L max av = average of minimum length, L max wm = weighted mean of maximum length, Wmin wm = weighted mean of minimum recorded width, Wmin av = average of minimum width, Wmax wm = weighted mean of maximum recorded width, Wmax av = average of maximum width, Tot. Meas. = total number of eggs measured for weighted mean estimates, Refs wm = total number of references for weighted mean estimates, Refs all = total number of references for overall averages.

in the review [58]; or where localities were referenced in another publication, but the primary text was unavailable [59–62]. If the same occurrence was referenced in separate papers, this was recorded. Any occurrences additional to the main data in a study (separate surveys, case reports or personal communications) if not reported in a separate paper, were recorded, with the publication in which they were included/cited as the relevant reference, e.g., Richard Bradbury pers. comm. in Morter et al. [63].

**Georeferencing.** Coordinates included in a publication were recorded and transformed to decimal degrees. Non-georeferenced localities were located with Google Maps or the place-names database geonames.org, and the data source noted. If the locality name was not found, gazetteers were checked at the NHM library, and paper maps were scanned and checked for site names and geographical details. In some cases localities could not be georeferenced. Where localities or endemic zones were plotted on site maps in a publication, they were mapped using the GDAL georeferencer for QGIS, Version 3.10, Coruña (QGIS.org, 2020), and collated. In Voelker & Sachs [44], many crab occurrences were included in 4 published maps, all georeferenced using GDAL, and site names and locations on maps cross-referenced with Google Maps. Post georeferencing, current region and district names were recorded as well as verbatim ones given in publication, by a spatial join in R to records from GADM version 3.6, the database of GADM database of Global Administrative Areas accessed through the GADM portal at https://gadm.org/.

**Uncertainty assessment.** Many records, historical and recent, had a high degree of coordinate uncertainty. Contextual notes were made, and transformed to a category to attempt to assess uncertainty in a consistent and semi-structured way. For example, a case report where the patient lived in an endemic disease focus and had never lived elsewhere would be assessed as most likely a case of local transmission and a low uncertainty; vice versa for a patient who had also travelled to different regions and therefore could have obtained the infection elsewhere. Uncertainty was also assessed by site category. Hospitals were recorded as high uncertainty as patients in endemic regions may have travelled considerable distances to the nearest clinic or hospital, far from the site of transmission [19]. If the village/town that a patient was from was not recorded, the hospital locality was used. Schools were recorded as low uncertainty, with the balance of probability that the transmission site was local and that school children were unlikely to have lived previously in other endemic regions. For crab collection sites, markets were assessed as low uncertainty for those in rainforest zones where crabs are generally sold live and fished locally, within a 20 km radius for example (NC, pers. comm.). In urban markets however, crabs may be transported, sometimes even across borders [64], and these were recorded as high uncertainty. Direct crab collection sites, i.e., crab fishing sites in rivers/streams, were recorded as low uncertainty, as this represents direct sampling of the crab habitat; crabs do not move far along a watercourse, reproduce by direct development and have a small home range, although limited seasonal movements may occur with high water conditions (NC, pers. comm.). Mammal survey sites included forest regions, generally nearby settlements, or within settlements themselves for domestic animals. Both hospitals and crab markets were classified as 'indirect' sites, potentially representing a large catchment area, particularly hospitals. These low uncertainty localities were marked out separately on maps, e.g., *Paragonimus* life stages isolated from crab hosts or non-human mammal hosts (excluding domestic animals).

## Results

### Study selection

A total of 874 studies were initially recovered from databases, 169, Google scholar; 170, Pubmed; 64, Africa Journals Online; 362, Science Direct; 33, Web of Science; and an additional

76 from other sources, reference lists, and scanning journal holdings in NHM library (Fig 1). After removal of duplicates, 512 titles were screened, of which 205 full texts were identified, where 16 were unavailable and 39 were excluded with reasons given (S1 Table). This resulted in a final total of 143 studies selected for the systematic review, with all databases above represented apart from Science Direct (and all 76 publications from other sources, i.e. reference lists as above), and 7 for the subset review of *Poikilorchis congolensis*.

### General findings/Study characteristics

Of the 143 scientific publications selected, 131 were peer-reviewed publications from 72 journals, and the remainder consisted of 3 book chapters, 2 thesis copies, 3 conference papers, 3 publications from grey literature (government reports/publications), and 1 manuscript in preparation. One journal was particularly strongly represented with 25 publications (Tropical Medicine and Parasitology, formerly Tropenmedizin und Parasitologie). For *Poikilorchis congolensis*, 7 publications were selected, all published in peer-reviewed journals.

### Species

The numbers of studies by species were as follows; *Paragonimus uterobilateralis* (44), *P. africanus* (15), both *P. africanus* and *P. uterobilateralis* (8), and 1 each for *P. kerberti* and *P. gondwanensis*. For 69 studies, the species was recorded here as *Paragonimus* sp. because the species was either not stated or incorrectly recorded as *P. westermani*. Two surveys found no evidence of *Paragonimus*, but were included as absence data [63,65]. The remaining 3 publications were recorded as Trematoda as were solely observations of microcercous cercariae in snails, regarded as unconfirmed [30,31,36; S1 Table]. Positive molecular identification was not a suitable criterion for exclusion having been utilised in only 2 studies [29,39]. A study that reported parasite eggs morphologically consistent with *Paragonimus*, but with PCR amplification unsuccessful was therefore considered a confirmed case for this review [66]. This author also retrospectively evaluated an earlier survey with an assessment of "*Paragonimus*-like" infections as positive for *Paragonimus* based on the available evidence [67], we concurred with the assessment. For *Poikilorchis congolensis*, 1 case was diagnosed as caused by either *Poikilorchis* or *Paragonimus* [68]. With symptoms of a retroauricular cyst containing *Poikilorchis* eggs and absence of eggs in sputum, consistent with *Poikilorchis* only; the case was assessed to be the latter (the study authors made a similar observation). Some early publications mistakenly noted *Poikilorchis* cases as *Paragonimus*, but these were corrected in later publications for the same reason (*Poikilorchis* symptoms present only), and the same assessment was made here.

In total, the publications reported ca. 6400 cases of infection with *Paragonimus* spp. (freshwater crab hosts and human and mammalian hosts together across studies), and ca. 4800 cases in humans only. For *Poikilorchis congolensis*, 7 publications recorded 11 cases in total. 11 cases based solely on serology, and 3 on symptoms only were recorded as unconfirmed and excluded from totals. Mixed infections were reported from crabs in one study, also from Cameroon [44]. No reports of mixed infection in humans were found. One study from Southwest Cameroon reported eggs of *P. africanus* and *P. uterobilateralis* from patients, it was not recorded if any patient was expelling both species [69].

### Locations

The majority of studies were from 4 countries: Nigeria, 53; Cameroon, 27; Liberia, 14; and Côte d'Ivoire, 11 (Fig 2). *Paragonimus africanus* was recorded from Cameroon, DRC, Equatorial Guinea, Gabon and Nigeria; *P. uterobilateralis* from Cameroon, Gabon, Guinea, Liberia

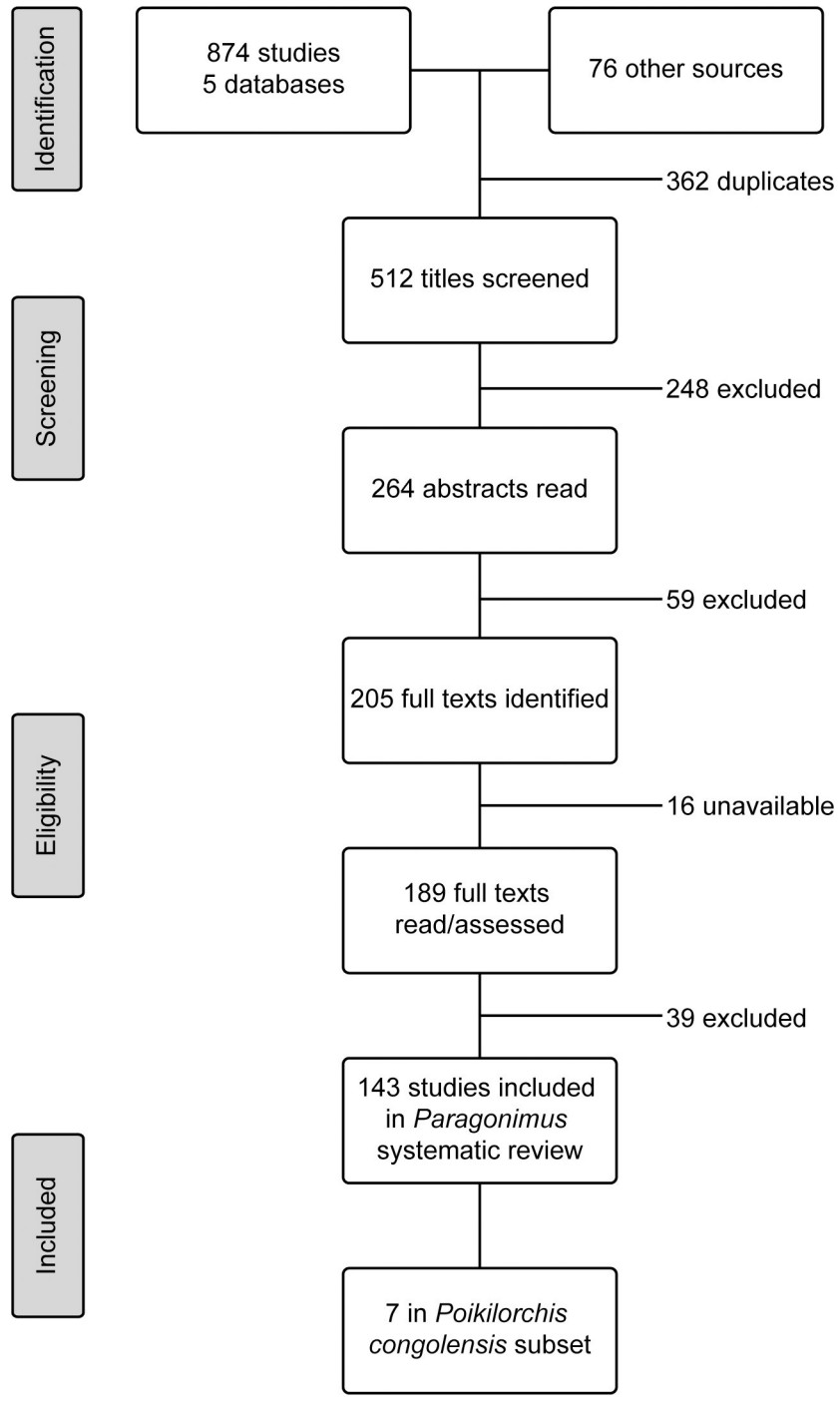

**Fig 1. Flow diagram for the selection and assessment of studies for the systematic review of paragonimiasis and *Paragonimus* spp. and the subset of *Poikilorchis congolensis*, in Sub-Saharan Africa.**

and Nigeria; *P. gondwanensis* and *P. kerberti* from Cameroon; and *Paragonimus* sp. (where species was not specified) from Benin, Côte d'Ivoire, Gambia, Ghana, Libya, Republic of Congo, Senegal, South Africa and Tanzania, in addition to previously mentioned countries Cameroon, DRC, Equatorial Guinea, Gabon, and Nigeria.

## Survey period

The systematic review included all available publications since *Paragonimus*-like lung infections were first reported in Africa, from 1920 to the present. The total number of studies by decade were as follows: 1910s (1), 1920s (1), 1930s (3), 1940s (3), 1950s (2), 1960s (8), 1970s (34), 1980s (27), 1990s (21), 2000s (16), and 2010s (26) (and an additional publication assessed data from 1970s-2000s). Publications increased substantially in the 1970s after the outbreak of paragonimiasis in Nigeria during the Biafran war, also known as the Nigerian Civil War [19; Fig 2]. The majority of recent publications were from Nigeria (Fig 2). For *Poikilorchis congolensis*, apart from one relatively recent case report in the United Kingdom from a Nigerian patient [70], the remaining 6 publications dated from 1943 to 1975.

## Study type

Literature by study type included: 31 case reports (sole human cases); 3 vet case reports (for sole domestic cat/dog cases); 11 case reviews (multiple cases, 3 also including surveys of crab intermediate hosts and 2 including clinical studies); 10 experimental studies; 8 clinical studies

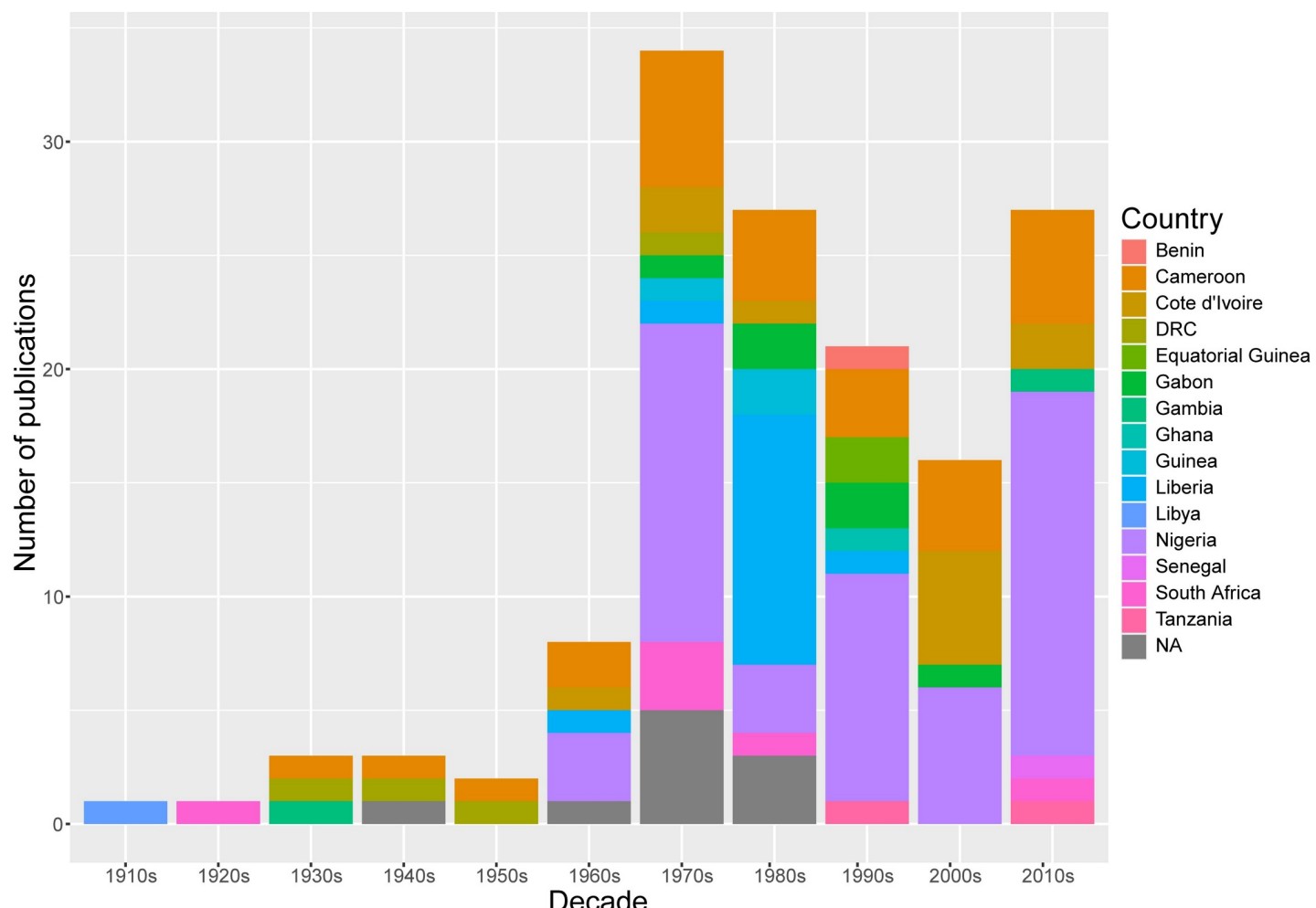

**Fig 2. Total number of studies by decade and country.** Publications of experimental studies (1970s and 1980s) or case reports where the origin of infection could not be ascertained (1940s, 1960s) are shown in grey (country = NA). For a longitudinal study, the final decade only is shown (87).

(including 3 drug treatment studies); 3 general biology studies, and 3 species descriptions. There were 70 surveys in total: 27 solely human surveys (several also including clinical studies/ drug treatment trials); 4 solely livestock surveys; 8 solely non-livestock mammal surveys; 3 solely snail surveys; 8 solely crab surveys; and the remainder of 20 various combinations of above survey categories. Experimental studies were carried out in the 1970s and 1980s (Fig 2), generally with mandrills, *Mandrillus leucophaeus* or rhesus macaques, *Macaca mulatta* as the experimental host [71]. An additional 3 surveys were questionnaire-based only, not included in further data analysis. For *P. congolensis*, apart from the species description [41], all publications were case reports or reviews (S1 Table).

## Subset analysis

**Prevalence and intensity of infection in humans.**   Prevalence estimates were included in 42 publications, of which 31 met our criteria for further analysis (S1 Table). In this subset, the study population size range was 127–3600 (average 1091). The total number of studies by country: Nigeria (18), Cameroon (10), Liberia (1), Equatorial Guinea (1), and Gabon (1). Between 3–10 studies were carried out per decade since the 1970s (none from 1960s), and 1 survey also from the 1950s [18]. The majority of studies were cross sectional; both schoolchildren and adults were surveyed, 7 surveyed school-aged populations only. In 10 studies other hosts were also surveyed e.g., crabs. Most studies surveyed populations within a given area, especially within known endemic regions, but many compared prevalence by locality within the surveyed region.

The overall prevalence in the subset (*Paragonimus uterobilateralis* and *P. africanus* or unspecified) ranged between 0–27.0% (average: 8.4%). Average prevalence by decade, for both species combined: 1950s, 4% from 1 study only, 1960s no data, 1970s, 5.2% (5 studies), 1980s 8% (3 studies), 1990s 7.6% (8 studies), 2000s, 7.7% (6 studies), and 2010s, 12.3% (8 studies). In total, 11 publications reported a study prevalence over 10%, 1 from Cameroon [72], and the remainder from Nigeria, and 6 from recent surveys, post 2000 [22–24,26,33,73–77].

Average prevalence across studies was significantly higher overall in *P. uterobilateralis*: at 12.4%, range 0–27% (1670/13,443, 12 studies), than *P. africanus*: at 2.9%, range 0.4–9.5% (263/ 8978, 8 studies, $P$ = 0.009, Table 1). The average prevalence for surveys where the species was not specified (*Paragonimus* sp.) was 8.7%, range 0.2–20.8% (992/11415, 7 studies from Nigeria, 3 from Cameroon). By country, prevalence of *P. uterobilateralis* for studies in Nigeria only at 12.5% (11 studies) was significantly higher than prevalence of *P. africanus* for studies in Cameroon only at 3.4% (7 studies, $P$ = 0.008).

For the intensity of infection subset (Tables 1 and S3), 14 studies reported intensity measures in human surveys. The overall intensity of infection for *P. uterobilateralis* was 108.5 e/5g, Table 1, 9 studies, all in Nigeria apart from one in Liberia, [78]; for *Paragonimus* sp. where the species was not specified, 10.8 e/5g (4 studies). No intensity of infection data for human surveys was available for *P. africanus*. Range was recorded for 5 studies (Nigeria only), overall, 1–903 e/5g. In the three studies where both sputum (e/5g) and stool (epg) counts were reported, stool showed lower egg counts overall [23,78,79]. In studies where intensity was assessed by category, 3 had standard categories (low at <50 e/5g, medium at 51–100 e/5g, and high at 101+ e/5g); where the overall averages were 43.3% for low intensity, 30.6% for medium and 25.9% for high intensity [33,73,80].

**Prevalence and intensity in other mammals.**   Mammal studies often involved sporadic or opportunistic sampling and very small sample sizes, with few systematic surveys [but see 39,66,67,81; S2 Table]. In total, 27 publications included data from mammals, of which 16 included viable prevalence data (publications excluded if the total surveyed was not given).

Surveys were generally carried out in forest near settlements [40] or in case of domestic animals, at settlements themselves [5,81]. Sampling generally involved collection of scats/faecal samples, but in some cases, sacrifice of mammal hosts and resulting autopsies [34,71]. Mammals surveyed included: primates (cercopithecids, lorids), Feliformia (viverrids, herpestids); domestic animals (cats, dogs and pigs); Eulipotyphla (*Crocidura*) and Rodentia (*Atherurus* and assorted Muridae). The overall prevalences in all non-human mammal hosts were as follows: *P. africanus*, range 20–100%, average 43.2% (35/81, 5 studies), *P. uterobilateralis*, range 12.5–100%, average 35.7% (46/228, 4 studies), and *Paragonimus* sp., range 14–100%, average 15.5% (56/362, 6 studies). Prevalences were as follows, in Feliformia: *P. africanus*, range 25–100%, average 58.8% (10/17, (*Civetticis civetta*, *Crossarchus obscurus*, *Nandinia binotata*, 3 studies), *P. uterobilateralis*, range 40–100%, average 77.8% (35/45, *Civetticis civetta*, *Crossarchus obscurus*, 4 studies, and *Paragonimus* sp., range 20–100%, average 25.8%, *Civetticis civetta*, *Crossarchus obscurus*, *Ichneumia albicauda*, *Atilax paludinosus*, 8/31, 3 studies); and in primates, *P. africanus*, range, 20–50%, average 39.1% (25/64, *Cercocebus torquatus*, *Mandrillus leucophaeus*, *Perodicticus potto*, 3 studies), and *Paragonimus* sp., range 14–14.6%, average 14.4% (47/327, *Papio anubis*, 2 studies, Table 1). Intensity was reported in 3 studies only, 96.7 epg of *P. africanus* in *Cercocebus torquatus* [39], 17 epg of *P. uterobilateralis* in *Civetticis civetta* [82], and *Paragonimus* sp. 47 epg in *Papio anubis* [66; Table 1].

**Prevalence and intensity in crabs.** Several systematic crab surveys undertaken near known foci of infection were reported in the literature. Considering parasitological screening methods, generally tissue was homogenised and washed, and resulting sediment and supernatant screened for metacercariae. In 1 study, only the haemocoel was screened [83]. The overall prevalences for all genera/species pooled in freshwater crabs were as follows: *Paragonimus africanus*: range 0–88% (average 19.9%), (103/333 crabs surveyed total, 4 studies), *P. uterobilateralis*: range 0–100% (average 25.1%) (542/8300 crabs surveyed total, 14 studies). Prevalence and intensity of *Paragonimus* spp. varied by genera and species of crab (Table 1). For the most common crab host, *Sudanonautes africanus*, prevalence pooled across studies for *P. africanus* was significantly higher at 42.9% (103/240, 4 studies) than for *P. uterobilateralis* at 5.9% (*P* = 0.031, 156/2634, 5 studies). Overall intensity of infection in *S. africanus* across studies was also significantly higher for *P. africanus* at 54.6 metacercariae per host (106 crabs total, 2 studies) than *P. uterobilateralis* at 6 (*P* = 0.002, 34 crabs total, 2 studies; Table 1). Prevalence for *P. kerberti* in *S. africanus* was high at 32.9% (493 crabs in total, 1 study). In *S. pelli*, prevalence was low for both *P. africanus* at 1.5% (1/66, 1 study), and *P. uterobilateralis* at 3% (2/66, 1 study), although data were only available from 1 study which included co-infections [44].

*Liberonautes* has been reported as a host only for *P. uterobilateralis* to date. *Liberonautes latidactylus* had both high prevalence of *P. uterobilateralis* (average 24.4%, 87/357, 4 studies) and intensity of infection (117.1, 85 crabs total, 3 studies, Table 1). Other species of *Liberonautes* where data were available showed low prevalence (and intensity) of infection, *L. chaperi* had a prevalence of 2.3% (1/4l, 1 study), and *L. nanoides*, 0.5% (4/753, 1 study; Table 1).

**Egg size measurements.** In total, 37 studies included egg size measurements for *Paragonimus* spp., and 6 for *Poikilorchis congolensis* (Tables 2 and S4). The overall averages and ranges for length and width were as follows (all hosts combined, weighted averages): *Paragonimus africanus*: length: average 91μm (+/-3.1 SD), range min and max averages, 76.4–107.5μm; width: average 49.5μm (+/-3.1 SD), range min and max averages, 43.6–59.1μm (from 2472 eggs, 9 studies); *Paragonimus uterobilateralis*: length: average 69.8μm (+/-4.1 SD), range min and max averages, 58.2–79.9μm; width: average, 41.1μm(+/-2.1 SD), range min and max averages, 36.2–46.1μm (from 3159 eggs, 9 studies); *Paragonimus* sp.: length: average: 88.2μm (+/-12.2SD), range min and max averages, 73.4–101.1μm; width: average, 52.7μm (+/-12 SD); range min and max averages, 43.1–60.1μm (from 472 eggs, 7 studies total, Fig 3). No data were

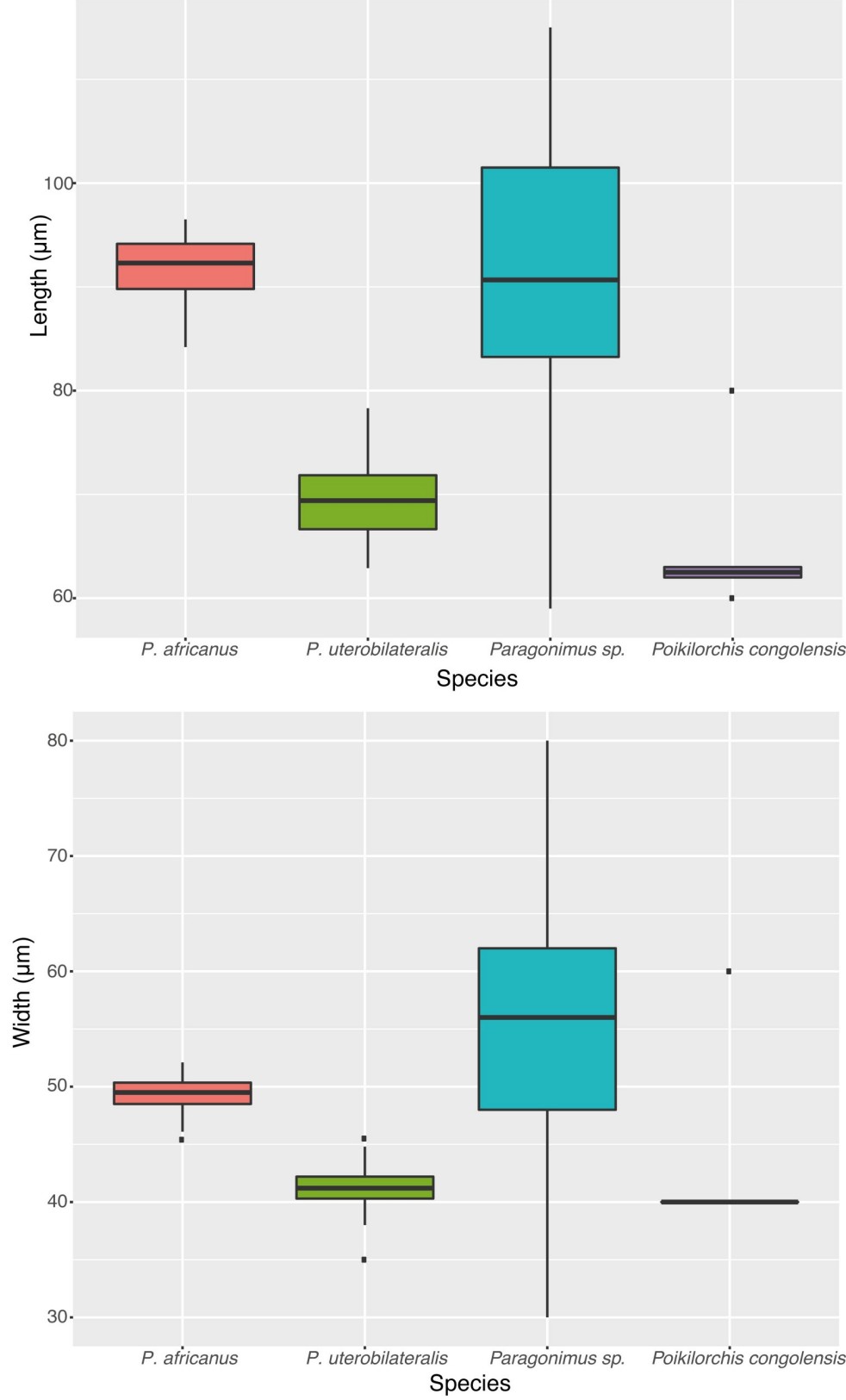

**Fig 3. Average P*aragonimus* spp. and *Poikilorchis congolensis* egg length and width measurements in µm reported in the literature, combined data from 37 studies in total.** Data from *P. gondwanensis* and *P. kerberti* data were not available therefore not shown.

available for either *P. gondwanensis* or *P. kerberti*. For *Poikilorchis congolensis*: average length was 65μm (+/-7.4 SD), average width 43.3μm (+/-8.2 SD) (non-weighted average, Table 2); and range was 60–63× 40μm for all eggs apart from one measurement, at 60 × 80μm [68, S4 Table]. Measurements were similar by host, i.e., eggs isolated from humans or other mammals showed similar values (Table 2). Metacercariae measurement sample sizes were very small with high standard deviations, but show in contrast to egg measurements, *P. uterobilateralis* metacercariae are generally larger sizes: *P. africanus* 564μm (+/-228.8 SD, from *Sudanonautes* spp., 3 studies), *P. uterobilateralis*, 701.6 (+/-119.4 SD, *Sudanonautes* and *Liberonautes*, 4 studies), and *Paragonimus* sp. (species not specified), 468.3 (+/-144.2 SD, *Sudanonautes*, *Liberonautes*, 2 studies, Table 2).

From the prevalence subset data, general qualitative observations were made for any prevalence trends evident by demographics (age and sex) for human surveys. Additional observations were made on clinical symptoms, drug treatments and other diseases surveyed where reported in any publication. These observations are summarised in the following sections.

**General epidemiology.** Most studies in the human prevalence subset analysis examined prevalence differences by age (21 studies) and sex (19 studies). Age categories were generally in 5- or 10-year divisions (e.g., 5–10, 11–15, 16–20 year olds), but varied by study population and publication (sometimes totals were not given or reported as region sub-totals). Because of this high heterogeneity, overall findings were aggregated to 10 year periods, and summarised as general observations (S2 Table). Highest prevalences were evident overall in younger age groups, where 10–20 year olds showed highest prevalence in 12 studies in total, including a longitudinal study which reported highest prevalences in this age category for 3 consecutive years [33]; followed by 1–10 year olds in 7 studies. For the remaining 2 studies, one categorised broad age ranges, reporting highest prevalence in 15–44 year olds [75], and the other reported highest prevalence in the 40+ age category [24]. Another study also reported second highest prevalence in 55–59 year olds [84]. Apart from these studies, a sharp decline in prevalence was evident for older age groups. Often prevalence levels between age categories were similar in the younger age groups, e.g., 5-10- and 11-20-year olds showing similar prevalences [22]. Other surveys not in the subset noted a majority of cases in patients younger than 20 years [85]. Examining differences in prevalence of infection by sex, 6 studies had higher prevalence in females than males, 10 studies, vice versa, and in 2, results were equivocal. Only 8 studies in total had significant differences, 3 with higher prevalence in females (all studies in Nigeria), and 5 with higher prevalence in males (4 studies in Nigeria, 1 in Cameroon). Unlike age, patterns of sex and prevalence of infection varied by study, and findings appeared to be largely study dependent, rather than indicating sex as a demographic factor affecting infection risk.

**Clinical Findings.** Clinical presentation of paragonimiasis was consistent across studies. Clinical signs and symptoms were mentioned in the majority of studies, and directly analysed or described in 18 publications, including 2 primate surveys [39,66]. The recorded symptoms in order of most to least common were: cough, hemoptysis, chest pain, abdominal pain, headache, fever, dyspnoea, fatigue, uritcaria, eosinophilia, and diarrhoea [23,28,29,33,80,86–91]. Hemoptysis was very common, with 1 study also reporting that periods of egg excretion in sputum correlated with hemoptysis [92]. Another study found no evidence of hemoptysis in any paragonimiasis case, although this was based on a small sample size [28]. In terms of unusual symptoms, 5 studies reported symptoms of epilepsy/convulsions/hemiplegia or similar in study participants positive for *Paragonimius* infection [12,18,29,90,91]. Six studies also detailed radiological findings, such as pulmonary "shadows"; lesions and cavitations [85, 86,88,91–93]. Two studies noted that it was difficult to distinguish the radiological patterning of paragonimiasis from early TB infection [85,86], although the latter study noted evidence of subcutaneous tissue wasting in study participants with TB only.

**Chemotherapies.** Several publications studied effects of drug treatment, including the following chemotherapies: Niclofolan [94,95]; Tricloabendazole [96], Bithionol [94], Menichlopholan [97]; and Praziquantel [69,79,98,99], with the study population treated ranging from 6 to 360. Early treatments included emetine and chloroquine, and appeared to have had low efficacy, required repeat treatments and had significant side effects [11,16,18,94]. Later treatments included Bithionol, Menichlopholan and Niclofolan, with higher success rates, but still evidence of side effects [94,97]. Praziquantel was first trialled in 1980 [98] and found to be highly efficacious with few side-effects [69], and was used exclusively from the 1980s onwards for treatment [76,86,100].

**Other diseases surveyed.** Nine studies surveyed and reported on TB infections, 3 with study subjects selected from those positive for pulmonary symptoms, and 3 with randomly selected study population (not showing symptoms). Studies that also screened for TB used the Ziehl Nielsen stain for Acid Fast Bacilli (AFB) for diagnosis. Two studies reported higher prevalence of TB than paragonimiasis, 19% vs 1.5% [101], and 16.9% vs 2.7% for [102]. The remaining studies had either higher prevalence of paragonimiasis than TB [21,27,86,87,103], or only paragonimiasis with no TB infections present [88,89]. In 3 studies, co-infections of TB and paragonimiasis were evident. In Sachs et al. [104] 1 case of coinfection in 3 paragonimiasis-positive patients was found (although only 2 were tested for TB), 3 cases in 100 for Nwokolo [21], and in Ibanga et al. [27] prevalences of TB only at 1.8%, and paragonimiasis at 9.4% including 10 patients with co-infection (0.9% of the total study population of 1100). There were also reports of patients with clinical histories of treatment for TB despite being tested as negative, and identified as positive for paragonimiasis in the present study [85]. Ten human surveys and 2 mammal surveys also screened for other parasites, generally soil transmitted helminths. In these studies, *Ascaris lumbricoides*, *Tricuris trichiura*, *Necator americana* and *Strongyloides stercoralis* (in decreasing order) were common; other parasites included *Schistosoma mansoni*, *Enterobius vermicularis*, *Entamoeba histolytica* and *Taenia saginata* [7,28,29,66,67,72,88,89,98,105–107].

## Mapping

For the mapping dataset, 132 publications were included (field surveys with viable occurrences/those including field-data), and for 120 publications at least 1 locality could be georeferenced (S5 Table). In total, 390 localities were selected for mapping (5 localities being *Poikilorchis congolensis* occurrences). Crab host genera show a wide distribution in West and Central Africa (*Sudanonautes* from Côte d'Ivoire to South Sudan, and *Liberonautes* from Senegal to Ghana, with occurrences spanning 1970–2017 (S1 Fig). The distribution of some of the freshwater crab host species (*Sudanonautes floweri*) extends beyond the known foci of paragonimiasis in the rainforests into the drier woodland and savannas of central and northern Nigeria, but only 2 outliers of *Paragonimus* reported from this region (Fig 4).

Examining *Paragonimus* distribution by species, *P. africanus* was reported mainly in Cameroon, but also in Nigeria in 2 studies [39,108], with 1 occurrence from the Cross River Basin, the other at Enugu, northwest of the endemic zone from a hospital patient [108]. In addition, sole reports were evident for DRC (no supporting data) [109]; Gabon (based on egg morphology) [110]; and Equatorial Guinea (also based on egg morphology, Fig 5) [111]. *Paragonimus uterobilateralis* in contrast was reported from Guinea to Gabon. Occurrences in outlier foci of KwaZulu-Natal in South Africa and Tanzania were reported as *Paragonimus* sp., although in Tanzania the infections, in olive baboons, *Papio anubis* were suspected to be *Paragonimus uterobilateralis* based on egg morphology [66]. Coordinate uncertainty was high for most localities, and those with low uncertainty occurrences (from non-human hosts, either crabs or

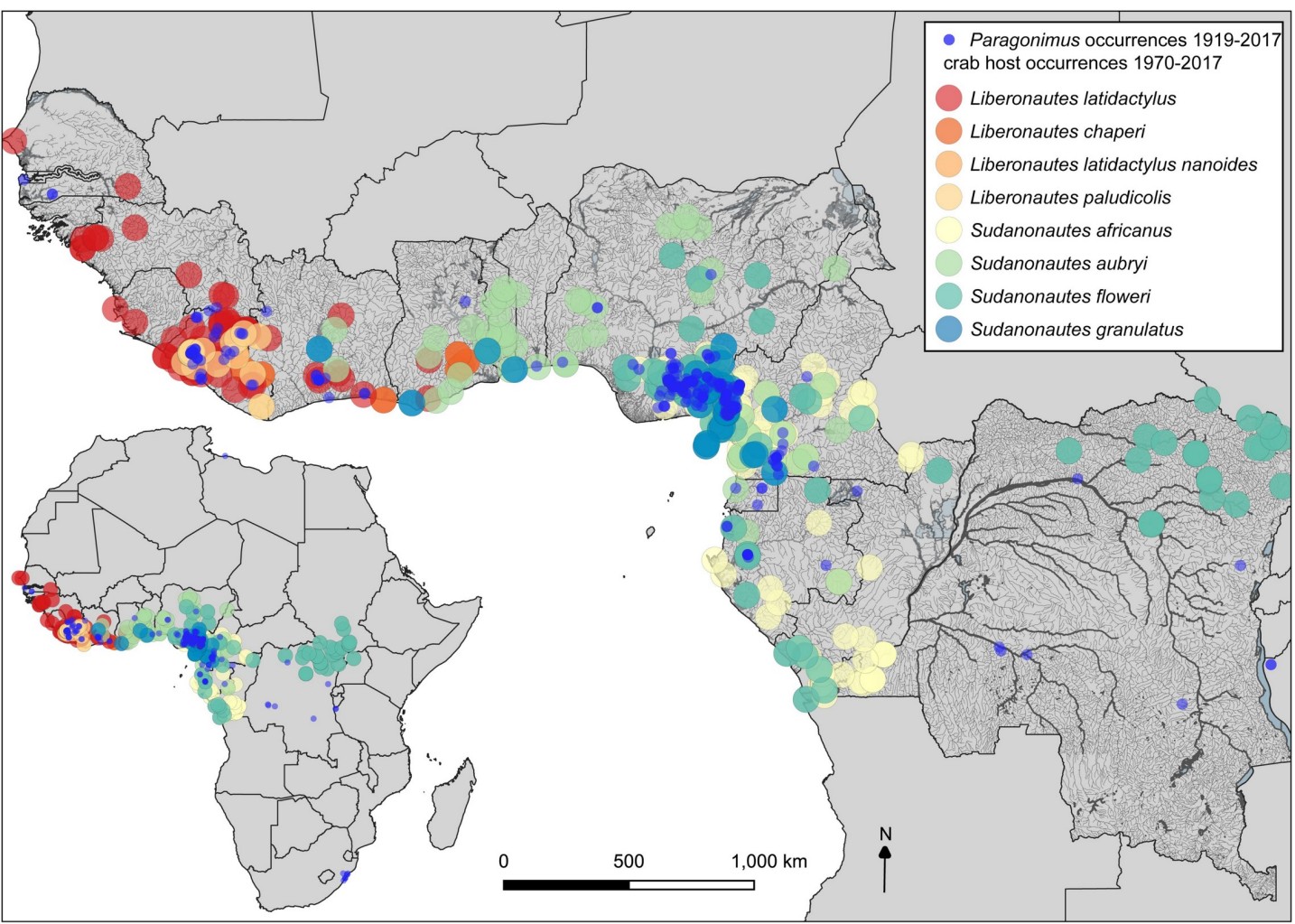

**Fig 4. Freshwater crab host occurrences by species.** 1970–2017, Africa wide, and West and Central African distributions. Crab occurrence data combines the dataset from author NC (S6 Table), and records from literature via systematic review (S5 Table). All river and waterway shapefile data sourced from DIVA GIS (downloaded from http://www.diva-gis.org/).

wild mammals), were all evident within the rainforest zone (Fig 5). There was evidence of temporal stability with occurrences spanning several decades at endemic foci at Southeast Nigeria and Southwest Cameroon (Fig 6). These main endemic regions also show the close association of these foci with the Cross and Imo River systems in the River basin/Gulf of Guinea region (Fig 5).

## Discussion

### Summary of key findings

In total, 143 papers were reviewed, where significant ongoing transmission of paragonimiasis with high prevalence and intensity of infection was uncovered, particularly for Southeast Nigeria. Across studies, we found higher prevalence overall for *P. uterobilateralis* than *P. africanus* in human surveys; variable prevalence and intensity of *Paragonimus* spp. infection in different crab hosts; and the potential endemicity of *P. africanus* in Côte d'Ivoire. In human surveys, the majority of the disease burden was consistently higher in younger age groups (<20 years).

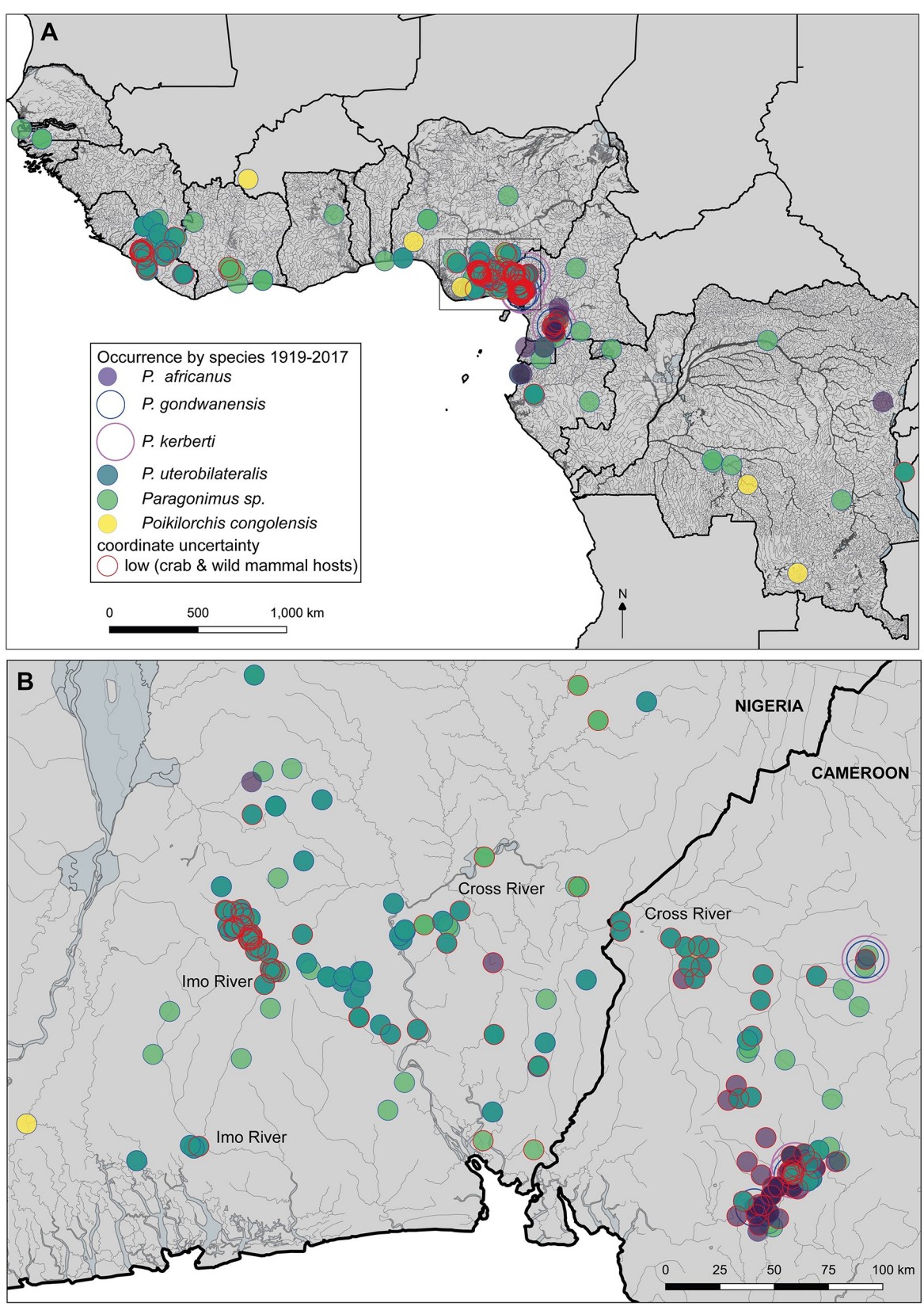

**Fig 5. Occurrences by species: *Paragonimus* spp. and *Poikilorchis congolensis* distribution from confirmed occurrences in the literature, 1910s- present (1919–2017).** (A), West and Central Africa, (B), close-up to the main endemic region, the southeast Nigeria/ Southwest Cameroon focus in the Cross River Basin, showing the Imo River and Cross River systems. Localities with low coordinate uncertainty (e.g. from intermediate crab hosts) are shown with red outlines.

This review also suggests paragonimiasis is a wildlife disease of importance in primates and Feliformia (viverrids and herpestids) in Africa. Mapping showed the close association of *Paragonimus* distribution with rainforest zones, but also outliers indicating localised transmission at locations with suitable ecology for all hosts. Temporal stability in endemic foci was evident. The mapping also illustrates extensive data gaps and under-sampling, and more widespread transmission than previously reported in Cumberlidge et al. [46].

## Infection in humans

We found numerous reports of significant recent transmission (1995–2016), with notably high prevalence and intensity in Southeast Nigeria in 15 studies, including 10 publications reporting prevalences over 10% [22–24,26,33,73–77]; Table 1). Recent transmission was also evident for Côte d'Ivoire (2006–2017) [40,53,112,113]; and Cameroon (1993–2014) [28,29,89,90,102].

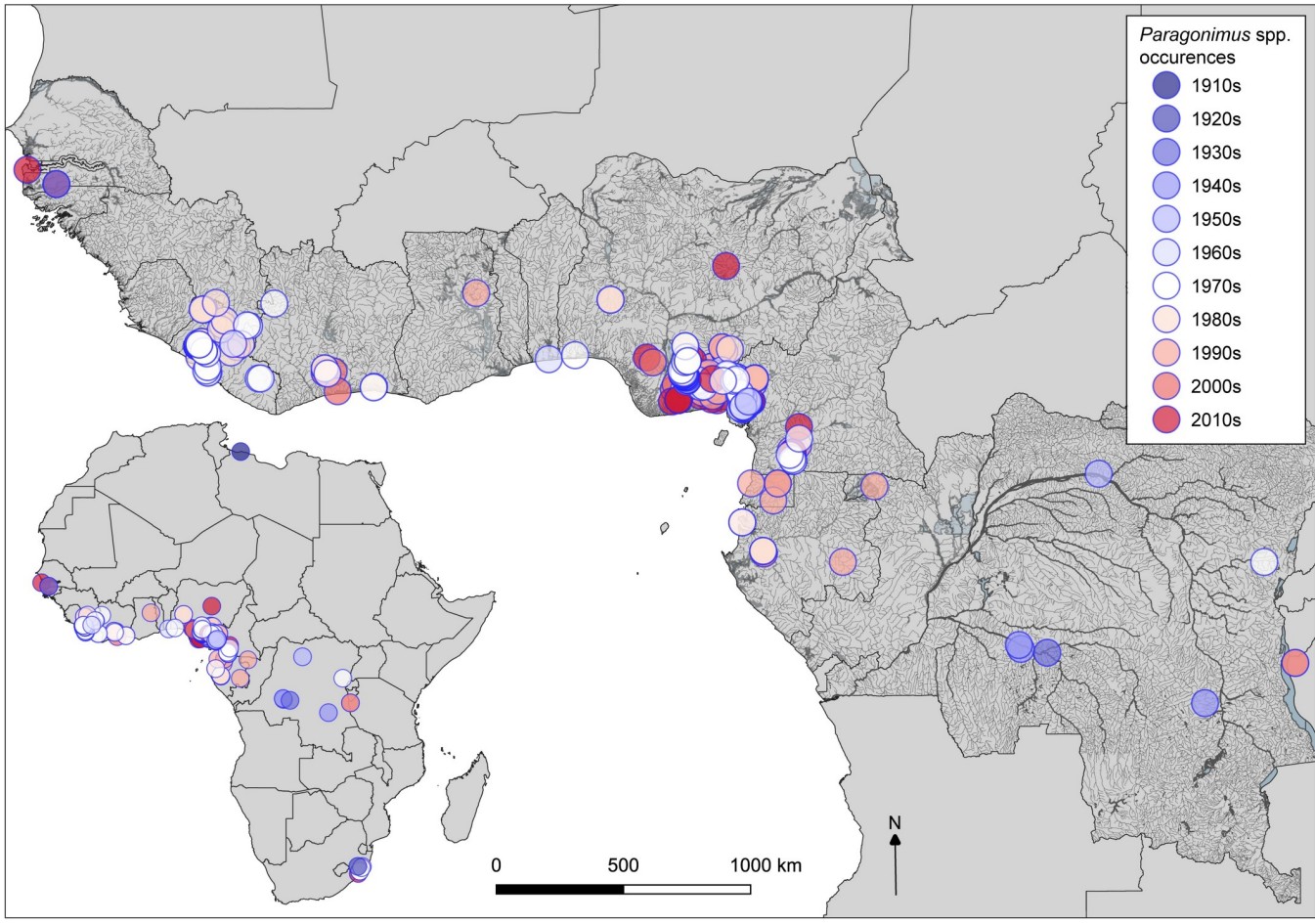

**Fig 6. *Paragonimus* occurrences by decade, 1910s- 2010s (1919–2017).** Main endemic foci in Southeast Nigeria, Southwest Cameroon show evidence of ongoing infection spanning decades. Another key focus of infection in Liberia has not been surveyed post the 1980s as shown. No recent case occurrences were evident in the literature for DRC and Republic of Congo, as shown.

Recent high prevalence in Nigeria has been attributed to widespread eating of uncooked freshwater crabs due to a lack of both current knowledge and health campaigns on the risks [24,114]. In rural communities in Nigeria, ongoing poverty may contribute to dependence on crabs as a food source; and inadequate sanitation in endemic regions is likely also to increase local transmission [23]. Prevalence in fact may be more widespread with under-reporting due to a lack of training of health workers to identify paragonimiasis [25,115,116]; TB surveys omitting differential diagnoses for paragonimiasis [83,117]; and reliance on microscopy for diagnosis (where low intensity cases may go undetected) rather than the more sensitive serology methods [29,117,118]. Serology is in fact a ubiquitous diagnostic for *Paragonimus* in Asia [2]. One relatively recent report showed evidence of co-infections of TB and paragonimiasis in Nigeria [27], supporting under-reporting by misdiagnosis as TB as a distinct possibility. Overall, these prevalence figures are closely associated with endemic foci, and given the highly focal nature of the disease should not be extrapolated nationwide [1]. Further, widespread praziquantel usage in many of the countries as chemotherapy for schistosomiasis may also be impacting paragonimiasis prevalence, although this has not been documented. However, the prevalence and intensity levels seen in these recent surveys in Africa are very troubling and would be cause for considerable attention in other less neglected tropical diseases [1].

Another unexpected result was the significantly higher overall prevalence of *P. uterobilateralis* at 12.4% than *P. africanus* at 2.9% ($P = 0.009$). For 11 surveys where the species identity was not reported, overall prevalence was 8.7%. As most of these studies took place in Nigeria, the majority were likely to be *P. uterobilateralis* (Fig 5), which fits the trend of higher prevalence of this species. This difference also reflects higher endemicity in Nigeria with overall prevalence of 12% (for *P. uterobilateralis*), vs *P. africanus* in Cameroon at 3.9%. Reasons for higher prevalence in *P. uterobilateralis* are unclear. *Paragonimus uterobilateralis* is much more broadly distributed, with a wider crab host range (Figs 4 and S1), but *P. africanus* may be present in other foci, and there may be inaccuracies in species identification in some studies. Molecular analyses are required to ascertain true distribution.

In other mammals, prevalence was variable and sampling generally sporadic with very small sample sizes, but prevalence was higher than in humans. Infections were primarily reported in Feliformia (viverrids and herpestids); primates (cercopithedids and lorids), and less commonly in domestic animals (cats, dogs and pigs). In Bakuza [66], olive baboons were observed eating freshwater crabs at Lake Tanganyika, identified as *Potamonautes* (although not screened for infection). The infected baboons showed evidence of advanced emaciation and high intensity of infection [119], suggesting a substantial worm burden. Friant et al. [39] found higher coughing frequency in infected *Cercocebus torquatus* in Nigeria, and suggested the primates may maintain local transmission as sylvatic hosts. The importance of paragonimiasis as a wildlife disease and the role of sylvatic hosts/degree of zoonotic transmission has not yet been fully realised. Further studies on the disease in mammals are needed using non-invasive faecal sampling [46].

Considering clinical findings and general epidemiology, a clear-cut association of higher prevalence with younger age groups was evident apart from two studies [24,84], where the timing of onset of infection would fit with those infected as children during the Biafran war. Five studies reported epilepsy/convulsive episodes/hemiplegia in paragonimiasis-positive patients, suggesting the possibility of extrapulmonary cerebral localisation [12,18,29,90,91]. In one of these studies, 2 cases were in fact assessed to be cerebral paragonimiasis [18]. Extrapulmonary (and cerebral) paragonimiasis has been frequently reported in Asian paragonimiasis [1,2], but appears to be rare in *Paragonimus* infections in Africa [48]. Nevertheless, recent reports of high rates of epilepsy in endemic regions [84] and incidences in paragonimiasis positive patients [29] requires further study.

## Infection in crabs

Both prevalence and intensity varied within genera according to *Paragonimus* species transmitted (Table 1). The most abundant host species *Sudanonautes africanus* had significantly higher prevalence ($P = 0.031$) and intensity ($P = 0.002$) of *P. africanus* than *P. uterobilateralis* infections overall. In *Liberonautes*, *L. latidactylis* showed high prevalence and intensity of *P. uterobilateralis* infections and in contrast *L. nanoides* very low levels of infection (Table 1) [120]. This presumably reflects susceptibility differences, which in turn would suggest crab species host range is a critical factor in *Paragonimus* distribution, although the key species have a wide range (Fig 4). Freshwater crabs are found throughout the range of *Paragonimus* in West and Central Africa, and it appears that the trematode is not host specific, developing in a number of crab species depending on which are found in the focus of disease [10]. The different species of *Paragonimus* may also adapt locally more to one host or another, resulting in local-scale differences in susceptibility. Data from co-infected *Sudanonautes africanus/pelli* in Mejek, Cameroon showed higher intensity of *P. africanus* within crabs in localities where it was more prevalent overall, likewise for *P. uterobilateralis* [44]. It is not clear what is driving the predominance of either species of *Paragonimus* in this region where they overlap, potentially an interaction of factors at a local scale. It would be worthwhile to investigate these processes further in mixed infection foci.

Another notable finding in the literature is evidence of up to 100% prevalence and high intensity in crabs within endemic foci, e.g., the Lower Bakossi, Cameroon [44]. This phenomenon has also been recorded at some foci for Asian *Paragonimus* [2]. We hypothesise that evidence of highly concentrated prevalence and intensity in crabs in endemic foci suggests accumulation of the parasites in the crab over time. This would fit with a food-borne transmission pathway in the crabs, whereby the infected freshwater/semi-terrestrial snail host is eaten. Indeed, there is evidence that *Sudanonautes* crabs in Nigeria feed on freshwater molluscs [121]. Alternatively, cercariae may leave an infected snail, either actively homing to a crab host, or adopting a sit-and-wait approach. For example, if snail hosts are freshwater-dwelling, once leaving the snail cercariae could disperse and detect freshwater crabs within waterbodies, or if the snail host is semi-aquatic, by crawling on damp vegetation. Waterborne cercariae are a common life history strategy in trematodes (e.g., *Opisthorchis*), enabling wide dispersal. The exact path of transmission at this point is not even known in the better studied Asian *Paragonimus* species however [1].

The role of crabs in epidemiology overall is key, as metacercariae can persist in crabs for years, and intensity of infection can determine infectivity [4]. There are however, few recent studies of *Paragonimus* infection in crabs [28,83,122], the last systematic studies being in the early 1990s [120,123–125]. Recent studies have suggested that parasite intensity has declined in crab hosts [28], but there are insufficient recent data to compare with the surveys of Voelker & Sachs for example [44]. Factors such as deforestation, hunting, pollution and waterway alterations are likely negatively impacting intermediate and definitive hosts of *Paragonimus*. Mammals in particular may have declined locally in areas under intensive hunting pressure [126]. However prevalences in humans do not show a decline and further investigation is needed to understand the overall picture.

## Egg/Metacercariae morphology and species distribution

Trends in egg size should be regarded with caution with potential misidentification, given overlap in egg size morphology; and that egg size can be affected by fixation/preparation method; and potentially by host [127]. For the identification of African *Paragonimus* species however, egg (or metacercariae) morphology is generally the only method or data available.

For Côte d'Ivoire, egg measurements of *Paragonimus* show a large size range (Fig 3, Tables 2 and S4); [128]. The variation may indicate the presence of both *P. africanus* and *P. uterobilateralis* in Côte d'Ivoire. Further, metacercariae with morphology consistent with *P. africanus* (i.e., thick-walled cysts) were evident from studies where the metacercariae were isolated from *Liberonautes/Potamonautes* (see Fig 5 in [40]). Both *Sudanonautes* and *Liberonautes* crab hosts are present in Côte d'Ivoire, the only country where they overlap and where *Paragonimus* has been found, apart from one livestock report from Ghana [81]. *Paragonimus africanus* has only previously been reported from Cameroon, Nigeria, DRC [109], Gabon [110] and Equatorial Guinea (Fig 5); [111]. Potentially *P. africanus* is present in Côte d'Ivoire, but alternatively it may be an undescribed species [40,129,130]. Other possibilities include a hybrid of *P. africanus* and *P. uterobilateralis*, or even a species complex (as may be the case for *P. westermani* in Asia) [1,2]. Overall Côte d'Ivoire presents a complex disease landscape. Here *Paragonimus* infections have also been reported in the brackish-water swimming crab *Callinectes* [53,64,112] but may be the same unidentified trematode in Traoré et al. [56]; (see Fig 35 in [131]; Fig 2 in [112]). Clearly molecular studies are required to clarify *Paragonimus* taxonomy in this region [40].

Unlike *Paragonimus*, *Poikilorchis congolensis* egg size showed little variation, almost uniformly 60–63 × 40um (Table 2). Potentially it also has a crab intermediate host, and there is evidence of overlapping distribution with paragonimiasis [132], but very little is known of general biology. Oyediran et al. [42] observed that the characteristic retroauricular cysts where eggs are effectively trapped could represent an evolutionary dead-end for the parasite in humans, although Schuster et al. [70] also found evidence of eggs in stool. Work is needed to ascertain the definitive host. If there are many questions for *Paragonimus* in Africa, there are many more for *Poikilorchis*. With benign symptoms and limited public health impact underreporting may be high, particularly given the main endemic region appears to be DRC; but it does seem to be a rare parasite. It is however certainly worthy of separate investigation.

## Distribution mapping

Mapping revealed few, but widespread outliers. Most occurrences were closely associated with endemic regions in rainforest zones [133] particularly low-uncertainty ones (Fig 5), illustrating the key importance of rainforest cover. However outliers also suggest localised transmission where ecology is favourable, such as cases in South Senegal and Gambia, although crabs have yet to be surveyed in this region. The main endemic foci in Nigeria and Cameroon are closely associated with the Imo and Cross River catchments (Fig 5). Endemic zones essentially represent regions of overlapping host distribution, with suitable ecological conditions for all hosts. Further studies on the ecology of the catchments are required to understand the disease landscape in this region. Although freshwater crab intermediate hosts are present in the Niger River catchment and Southwest Nigeria (Fig 4; S1 Fig), only 1 study has found infected crabs in these areas [134]. Crab hosts are also present in northern Nigeria, but only one report of *Paragonimus* (in livestock) evident [135]. Here limiting factors may be snail host distribution; or given that these regions are outside of the main rainforest zone, mammal host abundances may be low. Endemic foci also show temporal stability (Fig 6), similar to Asian *Paragonimus* species, e.g., at Sin Ho in Vietnam [2].

*Paragonimus uterobilateralis* showed a more widespread distribution overall, *P. africanus* in contrast more geographically contained, within tightly clustered endemic foci (Fig 5). Liberia showed more dispersed occurrences, due potentially to its more extensive inland forest [133]. Gaps were evident, with no records between Nigeria and Côte d'Ivoire, apart from Ghana [81], but in a domestic pig, and a case in Benin, but potentially acquired via market crabs from Nigeria [64]. These outlier cases should be confirmed with further studies. Likely ecological

factors are in play with the markedly different ecology of the Dahomey gap in the region, with much drier conditions and forests fragmented with savannah [136]. However, this could be partly due to sampling gaps, illustrated overall in the mapping (Figs 4–6). DRC for example has the most extensive rainforest in Africa, but sparse occurrences have been recorded, spanning an immense area. Moving further east, evidence of *Paragonimus* in primates in Tanzania [66] suggests potential for contiguous distribution from the Central African forests at least to Lake Tanganyika. Additionally, there is a focus in KwaZulu-Natal, South Africa [47]. Here potential freshwater crab hosts belong to *Potamonautes* [137], but no studies have screened these or any potential intermediate hosts for *Paragonimus* (Chris Appleton, pers. comm.). Little is known of the distribution of *Paragonimus* outside of the recognised endemic regions. *Paragonimus* spp. could be distributed far more widely than the known foci in West and Central Africa.

## First intermediate host

One of the more intriguing aspects of paragonimiasis in Africa is that the first intermediate snail host is yet to be identified [46]. Studies investigating snail hosts so far include the following as potential species: *Afropomus balanoidea* [34], *Homorus* (*Striosubulina*) *striateila* [35,36], *Potadoma freethii* [26,31,32], *Potadoma sanctipauli* [34], Achatinidae [30], and *Achatina achatina* (Roger Moyou-Somo pers. comm.). Some of these studies may have actually identified microcercous cercariae of *Paragonimus* in molluscs [30,35], but morphological and molecular data in combination are required for confirmation. As Asian *Paragonimus* has a wide host range within Rissooidea and Cerithioidea [9], African species may also infect a range of snails, contrary to the suggestion of Cumberlidge et al. [46] of a specific snail host range. The authors also proposed the first intermediate host snail may be amphibious/semi-terrestrial rather than fully aquatic, inhabiting the moist conditions of the tropical forest floor and picking up miracidia from a scat, in turn consumed by a semi-terrestrial freshwater crab, thereby infecting the second intermediate host. Transmission may even occur in both freshwater and semi-terrestrial snail species, especially given a likely wide snail host range.

## Limitations of study

Overall, reported prevalences are potentially underestimates, partly because they are based on microscopy only which lacks sensitivity. Improved diagnostics will likely lead to evidence of greater numbers of cases. However, the high prevalences reported in recent surveys also reflects a sampling bias toward high endemicity areas. Intensity in crabs is complicated by age (size) of crabs, with studies showing higher intensity in larger crabs [44,108,138], but there were insufficient individual measurements for further analysis. Existing bias within studies includes possibility of species misidentification; therefore higher prevalence of *P. uterobilateralis* would need to be confirmed with longitudinal molecular studies. In terms of potential bias in our methods, we analysed *P. kerberti* and *P. gondwanensis* as separate species, although they should be regarded as currently unconfirmed. Another potential bias is our estimate of study period where not reported, and temporal inaccuracy when infections were acquired when onset of symptoms was not clear in case reports. This was mitigated by assessing time period by decade only. All these factors may affect the quantitative data presented here, which may result in cumulative potential bias in our analysis. Further, our quantitative analysis does not represent a meta-analysis, not feasible given the array of non-comparable methods and data gaps, and the quantitative findings should duly be regarded as requiring confirmation in additional studies. As a general point, conflict and political instability may have impacted survey coverage. Paragonimiasis is most likely present in Sierra Leone, although to date no surveys

have been carried out to our knowledge; likewise in the DRC, all occurrences obtained from historical case reports only; and no recent surveys have been done in Liberia. These inherent sampling gaps reflect the state of the published literature, and could have been mitigated by expanding searches into 'grey' literature such as national Ministries of Health records, potentially useful particularly for countries like DRC where paragonimiasis surveys have not been published. A further study limitation was use of Google Translate for a limited number of works, which may have limited our understanding of some non-English publications. Overall however, the risk of bias at the study selection stage was low. Our search included all languages and some of the contributions included here were challenging to obtain. Undoubtedly some publications with medical implications have remained overlooked until now. Here we present a near-comprehensive literature review of all publications to date on African *Paragonimus* that considerably expands and updates the recent study by Cumberlidge et al. [46].

## Conclusions and future directions

Our systematic review has strengthened the existing body of knowledge on paragonimiasis in West and Central Africa. In terms of future work, molecular studies are urgently required to address major knowledge gaps in general biology. Utilising molecular methods such as metabarcoding and environmental DNA sampling could also help narrow the field of potential first intermediate hosts, which could then be intensively examined with dissection as prevalences may be low in the snail host, as seen in schistosomiasis [139]. Metabarcoding of infected crabs could also provide an approach to investigate potential first intermediate hosts via identification of diet items of the host crab species, this being a potential transmission pathway. In terms of mapping and spatial analysis, bayesian spatial modelling could illuminate potential *Paragonimus* distribution outside key endemic zones. Extensive human surveys in endemic foci, aided by better diagnostics, are also needed to establish the true extent of human infection, and treatment and control need greater consideration. As paragonimiasis is a foodborne disease, much could be done to promote safer eating practices. In the wake of the COVID-19 crisis, there is a renewed focus on zoonotic transmission and the One Health approach. Paragonimiasis-endemic regions, such as Cameroon, Nigeria and Côte d'Ivoire have a public health landscape of risk of infection with multiple NTDs as well as infectious diseases like malaria, TB and others. To achieve goals on reducing transmission of the NTDs, particularly in a One Health context, there is a need to understand all these diseases collectively. This is a call to action for the international NTD community to commit programmes of research into paragonimiasis in Sub-Saharan Africa.

## Supporting information

**S1 Fig. Freshwater Crab host occurrences by genus, 1970–2017, from Côte d'Ivoire to South Sudan, and *Liberonautes* from Senegal to Ghana.** Data compiled from the literature (S5 Table) and an the occurrence dataset from NC (S6 Table).
(TIF)

**S1 Table. Details for publications included in the systematic review (e.g. type of study, country where study carried out, brief description and contextual notes).** Full list of all publications where the full text was assessed, including those excluded from the final list with reasons given, is provided.
(XLSX)

**S2 Table. Prevalence data for publications included in prevalence analysis for all hosts, humans, other mammals, and intermediate host crabs (detailed in Materials and**

**methods).**
(XLSX)

**S3 Table. Intensity data for publications included in the intensity analysis for all hosts, humans, other mammals, and intermediate host crabs (detailed in Materials and methods).**
(XLSX)

**S4 Table. Egg measurement data for all reported measurements in the literature included in the systematic review.**
(XLSX)

**S5 Table. Occurrences (each unique sampling event/site record) from the literature and related data (detailed in Materials and methods).** Includes all georeferenced records of *Paragonimus* spp., and absence data (records where infection surveyed for but not present).
(XLSX)

**S6 Table. Previously unpublished geolocated records of occurrence for the freshwater crab second intermediate hosts based on empirically collected data from field research carried out by author NC from 1980 to 1984 in Nigeria and Cameroon, and from 1988 to 1989 in Liberia.**
(XLSX)

**S1 File. PRISMA (Preferred Reporting Items for Systematic Reviews and Meta-Analyses) checklist.**
(DOC)

## Acknowledgments

The authors are immensely grateful to the following people for their assistance: Rosie Jones, John Rose, Angela Thresher, Ben Nathan, Helen Pethers and Nial Briggs at the NHM library for assistance with interlibrary loans and scanning maps; authors David Aka, David Blair and Chris Appleton for sending publications; Chiho Ikebe for translation of Yamamoto et al. (1979), and authors Agathe Nkouawa, Richard Bradbury, Fabrice Zobel Cheuyem Lekeumo, Basil Okeahialam, Saffiatoe Darboe and Roger Moyou-Somo for sending additional data from published works.

## Author Contributions

**Conceptualization:** Muriel Rabone, David Rollinson, Neil Cumberlidge, Aidan M. Emery.

**Data curation:** Muriel Rabone, Joris Wiethase.

**Formal analysis:** Muriel Rabone, Joris Wiethase.

**Investigation:** Muriel Rabone, Joris Wiethase.

**Methodology:** Muriel Rabone, Joris Wiethase, Neil Cumberlidge.

**Validation:** Muriel Rabone, Joris Wiethase.

**Visualization:** Muriel Rabone.

**Writing – original draft:** Muriel Rabone.

**Writing – review & editing:** Muriel Rabone, Joris Wiethase, Paul F. Clark, David Rollinson, Neil Cumberlidge, Aidan M. Emery.

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
