## [Decision Letter · Decision Letter 0]

14 Oct 2020

Dear Rabone,

Thank you very much for submitting your manuscript "Endemicity of Paragonimus and paragonimiasis in West and Central Africa, a systematic review and mapping reveals stability of transmission in endemic foci for a multi-host parasite system" for consideration at PLOS Neglected Tropical Diseases. As with all papers reviewed by the journal, your manuscript was reviewed by members of the editorial board and by several independent reviewers. The reviewers appreciated the attention to an important topic. Based on the reviews, we are likely to accept this manuscript for publication, providing that you modify the manuscript according to the review recommendations. 

All four reviewers were favorably impressed by the paper. All indicated they wanted minor revisions. However, I note that reviewer number one did request quite a lot of rewriting. 

A few comments from myself. The geographical scope of the review is stated to be West and Central Africa. However, the rather small number of records from South Africa and e.g. Tanzania have been mentioned in the text. I wonder if the scope of the review should extend to all of Sub-Saharan Africa? Or even just "Africa", given that a record from Libya is mentioned.

In lines 41 and 71, "systemic" should read "systematic".

Line 771. How certain are you that the report of Paragonimus from pigs in Ghana is credible? The identification was based only of the presence of trematode eggs in pig feces. 

Line 829. Check spelling of snail taxon names.

As one of the reviewers pointed out, Scientific names must be in italics, That includes in the reference list!

Sincerely,

David Blair

Associate Editor

Banchob Sripa

Deputy Editor

All four reviewers were favorably impressed by the paper. All indicated they wanted minor revisions. However, I note that reviewer number one did request quite a lot of rewriting. 

A few comments from myself. The geographical scope of the review is stated to be West and Central Africa. However, the rather small number of records from South Africa and e.g. Tanzania have been mentioned in the text. I wonder if the scope of the review should extend to all of Sub-Saharan Africa? Or even just "Africa", given that a record from Libya is mentioned.

In lines 41 and 71, "systemic" should read "systematic".

Line 771. How certain are you that the report of Paragonimus from pigs in Ghana is credible? The identification was based only of the presence of trematode eggs in pig feces. 

Line 829. Check spelling of snail taxon names.

As one of the reviewers pointed out, Scientific names must be in italics, That includes in the reference list!

Reviewer's Responses to Questions

**Key Review Criteria Required for Acceptance?**

**Methods**

-Are the objectives of the study clearly articulated with a clear testable hypothesis stated?

-Is the study design appropriate to address the stated objectives?

-Is the population clearly described and appropriate for the hypothesis being tested?

-Is the sample size sufficient to ensure adequate power to address the hypothesis being tested?

-Were correct statistical analysis used to support conclusions?

-Are there concerns about ethical or regulatory requirements being met?

Reviewer #1: The study objectives and design are clear and well laid out. The design of the study did not require sample size estimation, although a seemingly sizeable number of articles have been collected and included in this review. Minimal statistical manipulation has been reported or presented and generally falls within the scope of descriptive. 

I question the legitimacy and validity of Google translation services as a robust tool for the translation of articles not presented in English, and lacking a knowledgable translator. For many languages, particularly of marginalised populations, this is not a reproducible or reliable translation tool.

A number of sections in the methods seem to be reporting results/ findings from the preliminary literature search and even the contents thereof. Authors should consider streamlining the presentation of methods alone and leave presentation of results to the appropriate section, in a systematic manner consistent with the methodology. Authors should consider presenting the methods as concisely as possible (probably with mention of deviations from the standard approach) and avoid including results or explanations thereof (discussions) within the methods section.

It is my opinion that following PRISMA guidelines, the authors have declared sufficient detail on selection and inclusion criteria. There is a bit of confusion in the statement/ explanation of the inclusion and exclusion criteria of articles, but this is mostly a problem of presentation.

The limitations of the study mentioned however, appear to apply more to the study reports reviewed and not specifically apply to this review. Key biases like volume and density of reporting, databases used to search for appropriate literature, and the inferences made from the meta-analysis done could improve the quality and confidence in the findings of this study

Reviewer #2: The objectives are clearly articulated with a clear testable hypothesis. The geographical coverage and the sample size is large which allows for the generation of robust scientific data.

Reviewer #3: **no**major new analyses/experiments require

Reviewer #4: -Are the objectives of the study clearly articulated with a clear testable hypothesis stated? Yes, they are 

-Is the study design appropriate to address the stated objectives? Yes 

-Is the population clearly described and appropriate for the hypothesis being tested? NA

-Is the sample size sufficient to ensure adequate power to address the hypothesis being tested? NA

-Were correct statistical analysis used to support conclusions? Yes

-Are there concerns about ethical or regulatory requirements being met? No

**Results**

-Does the analysis presented match the analysis plan?

-Are the results clearly and completely presented?

-Are the figures (Tables, Images) of sufficient quality for clarity?

Reviewer #1: Analysis presented does meet the requirement stipulated in the design. Generally results seem to be over detailed and contain very lengthy narratives that are quite difficult to follow. Authors should consider using short clear statements and present related results/ metrics together.

Results appear to be jumbled up in presentation from as early as multiple sections of the methods (239-262; 289-292; 302-310), as well as content more appropriate to the discussion or introduction (e.g. lines 197 - 206). Also, discussion of results should be left to the appropriate section as this makes the narrative lengthy and difficult to follow.

Some critical results e.g. common clinical findings/ presentations of paragonimiasis do warrant clear and independent presentation e.g. in a table, as this makes the information easier to appreciate. 

Some of the figures (e.g. Fig 3) could benefit from better descriptive legends and some of the key statistical information presented. 

Tables are quite detailed and presentable; therefore, some of the information presented in the results text is redundant and makes the paper somewhat over detailed and technical. Authors could consider watering down the results narrative to only key/critical findings, and refer readers to the tables for more detailed information.

Reviewer #2: The data has been well analysed using appropriate statistical test. Overall, the results are well presented in line with the specific objectives.

Reviewer #3: Table 1 appears to be missing

Reviewer #4: -Does the analysis presented match the analysis plan? Yes

-Are the results clearly and completely presented? Yes

-Are the figures (Tables, Images) of sufficient quality for clarity? Yes, they are

**Conclusions**

-Are the conclusions supported by the data presented?

-Are the limitations of analysis clearly described?

-Do the authors discuss how these data can be helpful to advance our understanding of the topic under study?

-Is public health relevance addressed?

Reviewer #1: Some confusion/ misrepresentation of discussion/ result material within the conclusion section (868-880). Overall conclusions are based on key findings from the study and provide useful insights into the future considerations and implications of the findings.

Good emphasis of public health relevance as well as policy and intervention plans are laid out in the conclusion section of the paper

Reviewer #2: The conclusions are accurately supported by the data generated.

Reviewer #3: The conclusions are clearly supported with the data and the authors describe the limitations of their data set. The significance of their findings to understanding the African paragonimiasis and public health relevance us adequately adressed.

Reviewer #4: -Are the conclusions supported by the data presented? Yes

-Are the limitations of analysis clearly described? Yes

-Do the authors discuss how these data can be helpful to advance our understanding of the topic under study? Yes they do

-Is public health relevance addressed? Yes it is

**Editorial and Data Presentation Modifications?**

Reviewer #1: The paper is generally of significant importance to the field and addresses a highly neglected trematodiasis on the continent;

1. Authors need to revise the language and sentence structure in a number of places. Use short and clear statements/ sentences especially when presenting complex concepts.

2. Consistency in presentation e.g. of non-English terminologies (with correct spellings every time e.g line 582 and related references to hemoptysis; line 592 and further mentions of Bithionol). Rules of consistent italicising scientific/ classification names should also be revisited 

3. Some general restructure of the methods, results and discussion sections should help make the paper easier to read and especially highlighting of key points for non-technical audiences. 

4. Thorough review and revision of the references (noting repetitions, numbering, formatting of references). Since this is a review of the literature, the references section indeed counts as the raw data/ results from the study and should be robustly presented.

Reviewer #2: Results:

• Lines 408 – 411: The number of studies doesn't add up to the totals. If you add up 132, 3, 2, 3, 4, it gives 144 and not 142 scientific publications as stated. There is also a consensus with figure 1 that indicates 149 studies on paragonimiasis, were included. And if you add to the 7 articles on Poikilorchis congolensis, it gives 156.

• Lines 611 – 615: Correct the spellings of the following organisms: Trichuris trichiura, Strongyloides stercoralis, Enterobius vermicularis. Verify the spelling of all other scientific names in the manuscript.

• Figure 3: Include the units of measurement in the Y-axes of the two graphs.

• Figures 3 and 5: Kindly improve on the resolutions of these figures; the labels are barely visible.

• Tables 1 and 2: kindly move the keys to the bottom of each table.

Discussion:

Lines 680-682: Refs 104 also reports on paragonimiasis and tuberculosis co-infection.

Reviewer #3: Minor edits:

109: Space missing between 3 and decades

116: Do you mean transmission “within” Nigeria?

129: P. gondwanensis and P. keberti are mentioned within the introduction and within results, but little information is provided on these putative (?) species. It would be helpful to have a little more background on how these are differentiated from the two main species, africanus and uterobilateralis. Some discussion on the reasoning for differentiating these species and the validity of this distinction would be helpful in the discussion. 

150: A brief introduction to Poikilorchis congolensis as related to Paragonimus spp. within the introduction would be useful. It is not immediately clear why it is included.

166: It is very clear how the human data and intermediate host data were handled for prevalence, intensity, mapping etc., but there is limited information on how data from other mammalian hosts were handled. Where missing, please include details on when they were excluded from analysis and why, or when they were included and how?

169: Briefly describe or at least spell out the PRISMA guidelines.

202: The sentence beginning with “Symptomology” is long and somewhat confusing. Consider breaking this down into multiple sentences and referring specifically to P. congolensis for clarity.

220: Is this referring to S1 and S5 tables? If so, S5 should be re-named S2 since it is the second table referenced in the manuscript.

309: The meaning of “…the levels reported were assessed in general observations” is not clear

324: Site type: These describe human, crab, and snail sampling location. None of these seem to describe sites where samples form wild animals would have been obtained.

330: Occurrences of crab intermediate hosts: would be helpful to have a brief description of this unpublished dataset, including how it was constructed. For example, were these extracted from the literature, based on niche modelling, from empirically collected data?

337: Unclear what is meant by the “subsequent source”

484: Suggest re-wording to enhance clarity. Include % prevalence for africanus in addition to uterobilateralis. Specify what the 3.4% refers to for Cameroon and what comparison is being made to produce the p-value.

502: Some indication of variation in the 43.2% statistic would be informative (e.g. range or SD)

503: Provide incidence data from Feliformia and primates or remove.

544: Highlight here that no information was available for P. kerberti or gondwanensis if appropriate

705: How does emaciation and high intensity infection suggest zoonotic transmission? This link is unsubstantiated/ unclear.

Figure 5: It is hard to tell the difference between kerberti and gondwanensis within the map, since they can often only be seen by partial circles. I suggest making the circles different colors.

Reviewer #4: Line 38: Maybe instead of writing “Intermediate hosts: freshwater snails and crabs”, it was better just to write as in line 88, “primary snail host” since land snails were suspected to play a role by some authors; even though, this information is yet to be confirmed. 

Introduction: It seems very long. I suggest that the authors should delete some sentences and bring them into the discussion session.

**Summary and General Comments**

Reviewer #1: Paper is generally well presented and of relevance to the field of parasitology, One Health, public health and the NTDs in general.

The study also addresses a glaring gap in the knowledge and scholarship on this group of disease causing trematodes.

A few concerns remain about the presentation and legibility of the content (highlighted and annotated in reviewed manuscript attached). Careful attention to references and presentation thereof (central to a review of the literature), as well as standard norms of presenting scientific/ technical terminology.

Reviewer #2: Rabone et al. present a systematic review, of the endemicity of Paragonimus and paragonimiasis in West and Central Africa. The authors did a fantastic job of capturing the epidemiology of paragonimiasis in the human, animal population, as well as the intermediate host, freshwater crabs. The data generated is very robust and has the potentials to impact research in this field, an area that is considerably neglected by the scientific research community. The major strength is the study design and the timeliness of the report. The studyl limitations have been correctly acknowledged by the authors in the discussion. Generally, the manuscript is well presented and written in acceptable English. However, there are a few minor concerns that need attention before the paper is publishable.

Reviewer #3: The authors review African paragonimiasis, arguably one of the most neglected tropical diseases in Africa. The literature on African Paragonimiasis is sparse, highly variable in methods and measures, making it challenging to piece together a coherent understanding of the current state of our knowledge on African paragonimiasis and its significance to public health. The authors have done an excellent job synthesizing these data. Their mix of quantitative and qualitative analysis is very highly appropriate, maximizing our knowledge gained from a heterogenous and challenging dataset. Their analysis improves our understanding of the parasite’s life cycle, epidemiology, and public health significance. This is a much needed and extremely valuable review, and I commend the authors for taking on this task and for their thorough review and careful analysis of data.

Reviewer #4: (No Response)

PLOS authors have the option to publish the peer review history of their article (what does this mean?). If published, this will include your full peer review and any attached files.

Reviewer #1: No

Reviewer #2: Yes: Tebit Emmanuel Kwenti

Reviewer #3: Yes: Sagan Friant

Reviewer #4: No
---

## [Decision Letter · Decision Letter 1]

29 Dec 2020

Dear Rabone,

Thank you very much for submitting your manuscript "Endemicity of Paragonimus and paragonimiasis in Sub-Saharan Africa, a systematic review and mapping reveals stability of transmission in endemic foci for a multi-host parasite system" for consideration at PLOS Neglected Tropical Diseases. As with all papers reviewed by the journal, your manuscript was reviewed by members of the editorial board and by several independent reviewers. The reviewers appreciated the attention to an important topic. Based on the reviews, we are likely to accept this manuscript for publication, providing that you modify the manuscript according to the review recommendations. 

I have received reviews of the revised version of the paper from original reviewers #1 and #3. They indicate that the paper is now acceptable for publication. However, reviewer #1 has offered two small points in an annotated version of the MS that you might like to consider. Whether you make changes in the light of these comments is up to you, but I have recommended "minor revision" to give you that chance. The two points are (there is no need to send you the annotated MS):

Line 255, beginning of the "Data analysis and summary statistics" section. Could consider including these R sections/ special packages as references? And the overall R software package? In some circles it is considered as a computational language with copyright properties.

Line 296. "Cumberlidge et al . (Table 2; 46)". Add publication year.

Sincerely,

David Blair

Associate Editor

Banchob Sripa

Deputy Editor

I have received reviews of the revised version of the paper from original reviewers #1 and #3. They indicate that the paper is now acceptable for publication. However, reviewer #1 has offered two small points in an annotated version of the MS that you might like to consider. Whether you make changes in the light of these comments is up to you, but I have recommended "minor revision" to give you that chance. The two points are (there is no need to send you the annotated MS):

Line 255, beginning of the "Data analysis and summary statistics" section. Could consider including these R sections/ special packages as references? And the overall R software package? In some circles it is considered as a computational language with copyright properties.

Line 296. "Cumberlidge et al . (Table 2; 46)". Add publication year.

Reviewer's Responses to Questions

**Key Review Criteria Required for Acceptance?**

**Methods**

-Are the objectives of the study clearly articulated with a clear testable hypothesis stated?

-Is the study design appropriate to address the stated objectives?

-Is the population clearly described and appropriate for the hypothesis being tested?

-Is the sample size sufficient to ensure adequate power to address the hypothesis being tested?

-Were correct statistical analysis used to support conclusions?

-Are there concerns about ethical or regulatory requirements being met?

Reviewer #1: The study objectives, methodology and analysis are clearly laid out, substantiated and structured in a clear and easily read manner.

No queries on ethics and clear commentary on regulatory matters.

Reviewer #3: -Are the objectives of the study clearly articulated with a clear testable hypothesis stated? YES

-Is the study design appropriate to address the stated objectives? YES

-Is the population clearly described and appropriate for the hypothesis being tested? YES

-Is the sample size sufficient to ensure adequate power to address the hypothesis being tested? YES

-Were correct statistical analysis used to support conclusions? YES

-Are there concerns about ethical or regulatory requirements being met? NO

All points of confusion were well addressed in the revisions of this manuscript.

**Results**

-Does the analysis presented match the analysis plan?

-Are the results clearly and completely presented?

-Are the figures (Tables, Images) of sufficient quality for clarity?

Reviewer #1: Appropriate analysis, clear and concise results both in tables and figures as well as the accompanying text

Reviewer #3: -Does the analysis presented match the analysis plan? YES

-Are the results clearly and completely presented? YES

-Are the figures (Tables, Images) of sufficient quality for clarity? YES

The authors have included additional results in this revision that address minor concerns.

**Conclusions**

-Are the conclusions supported by the data presented?

-Are the limitations of analysis clearly described?

-Do the authors discuss how these data can be helpful to advance our understanding of the topic under study?

-Is public health relevance addressed?

Reviewer #1: Well supported conclusions with succinct synthesis and linkage to the literature as well as inferences from key findings.

Good flow and structure in the narrative now.

Reviewer #3: -Are the conclusions supported by the data presented? YES

-Are the limitations of analysis clearly described? YES

-Do the authors discuss how these data can be helpful to advance our understanding of the topic under study? YES

-Is public health relevance addressed? YES

**Editorial and Data Presentation Modifications?**

Reviewer #1: Accept

I have included comments on suggestions for reference clarification in the attached file, but these are minor and suggestive only. Authors may effect them or choose to retain current formatting. Either way they do not constitute requisite revisions or revert for further review.

Reviewer #3: None

**Summary and General Comments**

Reviewer #1: The authors have done a splendid job in refining and restructuring the manuscript after initial reviewer comments (I read both my own and other reviewer comments for the overall appreciation of edits done). 

The manuscript is much clearer, easier to read/ digest and generally clearly highlights key and critical findings that have been linked very well with a vital public health agenda.

Reviewer #3: The authors have thoughtfully considered and thoroughly addressed all minor concerns in the previous version of this manuscript. The new manuscript will make a valuable and much needed contribution to our understanding of Paragonimus in Africa.

PLOS authors have the option to publish the peer review history of their article (what does this mean?). If published, this will include your full peer review and any attached files.

Reviewer #1: No

Reviewer #3: Yes: Sagan Friant
---

## [Editor Report · Decision Letter 2]

9 Jan 2021

Dear Rabone,

We are pleased to inform you that your manuscript 'Endemicity of Paragonimus and paragonimiasis in Sub-Saharan Africa: a systematic review and mapping reveals stability of transmission in endemic foci for a multi-host parasite system' has been provisionally accepted for publication in PLOS Neglected Tropical Diseases.

Best regards,

David Blair

Associate Editor

Banchob Sripa

Deputy Editor

---

## [Editor Report · Acceptance letter]

26 Jan 2021

Dear Rabone,

We are delighted to inform you that your manuscript, "Endemicity of *Paragonimus* and paragonimiasis in Sub-Saharan Africa: a systematic review and mapping reveals stability of transmission in endemic foci for a multi-host parasite system," has been formally accepted for publication in PLOS Neglected Tropical Diseases.

Best regards,

Shaden Kamhawi

co-Editor-in-Chief

Paul Brindley

co-Editor-in-Chief
